# MAVS integrates glucose metabolism and RIG-I-like receptor signaling

Qiao-qiao He[1,8], Yu Huang[1,8], Longyu Nie[1,8], Sheng Ren[1], Gang Xu[1], Feiyan Deng[1], Zhikui Cheng[1], Qi Zuo[1], Lin Zhang[2,3], Huanhuan Cai[2,3], Qiming Wang [4], Fubing Wang[5], Hong Ren[6], Huan Yan [1], Ke Xu[1], Li Zhou [1], Mengji Lu[7], Zhibing Lu[2,3], Ying Zhu [1] ✉ & Shi Liu [1,2,4,5] ✉

MAVS is an adapter protein involved in RIG-I-like receptor (RLR) signaling in mitochondria, peroxisomes, and mitochondria-associated ER membranes (MAMs). However, the role of MAVS in glucose metabolism and RLR signaling cross-regulation and how these signaling pathways are coordinated among these organelles have not been defined. This study reports that RLR action drives a switch from glycolysis to the pentose phosphate pathway (PPP) and the hexosamine biosynthesis pathway (HBP) through MAVS. We show that peroxisomal MAVS is responsible for glucose flux shift into PPP and type III interferon (IFN) expression, whereas MAMs-located MAVS is responsible for glucose flux shift into HBP and type I IFN expression. Mechanistically, peroxisomal MAVS interacts with G6PD and the MAVS signalosome forms at peroxisomes by recruiting TNF receptor-associated factor 6 (TRAF6) and interferon regulatory factor 1 (IRF1). By contrast, MAMs-located MAVS interact with glutamine-fructose-6-phosphate transaminase, and the MAVS signalosome forms at MAMs by recruiting TRAF6 and TRAF2. Our findings suggest that MAVS mediates the interaction of RLR signaling and glucose metabolism.

Glucose is a primary source of cellular energy. After uptake by glucose transporter (GLUT), glucose moves through three distinctly metabolic pathways, including glycolysis, the pentose phosphate pathway (PPP), and the hexosamine biosynthesis pathway (HBP)[1]. Glucose is either imported to mitochondria where it enters the tricarboxylic acid (TCA) cycle or shifts to the "Warburg effect" when oxygen is unavailable[2]. Hexokinases (HKs) are the first rate-limiting enzyme in glucose metabolism that catalyzes the conversion of glucose to glucose-6-phosphate (G6P)[3]. PPP branches off from glycolysis at the first committed step of glucose metabolism, leading to the synthesis of ribonucleotides; it is also the primary source of NADPH[4].

Glucose-6-phosphate dehydrogenase (G6PD) is the first and rate-limiting enzyme of the PPP[5]. Compared with glycolysis and PPP, only 2%–5% of the glucose that enters cells is directed to the HBP[6,7]. The HBP component glutamine-fructose-6-phosphate transaminase (GFPT) is the rate-limiting enzyme of the HBP and catalyzes fructose-6-phosphate (F6P) to the HBP end-product uridine diphosphate N-acetylglucosamine (UDP-GlcNAc)[8].

The innate immune response is the first line of host defense against pathogen infection, initiated by recognizing pathogen-associated molecular patterns by pattern-recognition receptors, including RIG-I-like receptors (RLRs) and Toll-like receptors (TLRs) and

[1]State Key Laboratory of Virology, Modern Virology Research Center, Frontier Science Center for Immunology and Metabolism, College of Life Sciences, Wuhan University, Wuhan 430072, China. [2]Institute of Myocardial Injury and Repair, Wuhan University, Wuhan 430072, China. [3]Department of Cardiology, Zhongnan Hospital of Wuhan University, Wuhan 430072, China. [4]College of Bioscience and Biotechnology, Hunan Agricultural University, Changsha 410128, China. [5]Wuhan Research Center for Infectious Diseases and Cancer, Chinese Academy of Medical Sciences, Wuhan 430072, China. [6]Shanghai Children's Medical Center, Affiliated Hospital to Shanghai Jiao Tong University School of Medicine, Shanghai 200000, China. [7]Institute for Virology, University Hospital Essen, University of Duisburg-Essen, Essen 45122, Germany. [8]These authors contributed equally: Qiao-qiao He, Yu Huang, Longyu Nie. ✉e-mail: yingzhu@whu.edu.cn; liushi_liushi@whu.edu.cn

cytoplasmic DNA sensors[9,10]. For example, in response to RNA virus infection, RLRs engage an adapter protein called MAVS (also known as IPS-1, Cardif, or VISA), located on the peroxisomes, mitochondria, and mitochondria-associated endoplasmic reticulum membranes (MAMs)[11,12]. Aggregated MAVS recruits signaling molecules to form the MAVS signalosome, including TNFR1-associated death domain protein, TNF receptor-associated factor (TRAF), and MITA (also known as STING and ERIS)[13]. Then, the MAVS signalosome initiates the activation of two cytosolic kinases (TANK-binding kinase 1 and IκB kinase), and their downstream transcription factors interferon regulatory factor 1 (IRF1), IRF3, and nuclear factor κB (NF-κB)[14]. As a result, these transcription factors enter the nucleus and promote the expression of type I interferon (IFN), type III IFN, and proinflammatory factors[15].

Elevated energy metabolism is required to meet the demands of effective immune functions[16]. Several studies showed that glucose metabolism regulates the innate immune via MAVS[17–19]. Nevertheless, the role of MAVS in glucose metabolism has not been established. Here, we show that MAVS is essential for shifting glucose metabolism from glycolysis to PPP and HBP in response to RLR signaling. Further experiments demonstrate that MAVS associates with G6PD and TRAF6 on the peroxisomes during RNA virus infection, resulting in the initiation of the PPP and type III IFN production. MAVS associates with GFPT2, TRAF2, and TRAF6 on the MAMs during RNA virus infection, resulting in the activation of HBP metabolism and type I IFN. These findings suggest that MAVS

is the central scaffold that coordinates glucose metabolism and RLR signaling.

## Results

### RLR activation regulates glucose metabolism reprogramming via MAVS

To determine whether MAVS participates in regulating glucose metabolism, we employed the Mavs knockout (KO) (*Mavs*[−/−]) mouse model, two specific short hairpin RNAs (shRNAs) for MAVS, and confirmed *Mavs* deficiency and shRNAs efficiency (Supplementary Fig. 1a, b). ShRNA-MAVS#1 was selected for the experiments described below. First, wild-type (WT) bone-marrow-derived macrophages (BMDMs) or *Mavs*[−/−] BMDMs were infected with vesicular stomatitis virus (VSV), and we performed glucose levels analyses. As shown in Supplementary Fig. 1c, VSV-infected WT BMDMs but not *Mavs*[−/−] BMDMs displayed elevated glucose levels. Similar results were obtained in Sendai virus (SeV)-infected or poly(I:C)-treated BMDMs (Supplementary Fig. 1d). The extracellular acidification rate (ECAR) and oxygen consumption rate (OCR) assay indicated that MAVS knockdown produced a reduction in ECAR and OCR (Supplementary Fig. 1e, f). Conversely, MAVS overexpression increased the ECAR and OCR (Supplementary Fig. 1e, f). [13]C[6]-glucose tracing metabolomics was employed to elucidate the mechanism of MAVS in glucose metabolism reprogramming. We measured [13]C-glucose incorporation through glycolysis, the HBP, the TCA cycle, and the PPP (Fig. 1a). VSV-infected WT BMDMs displayed

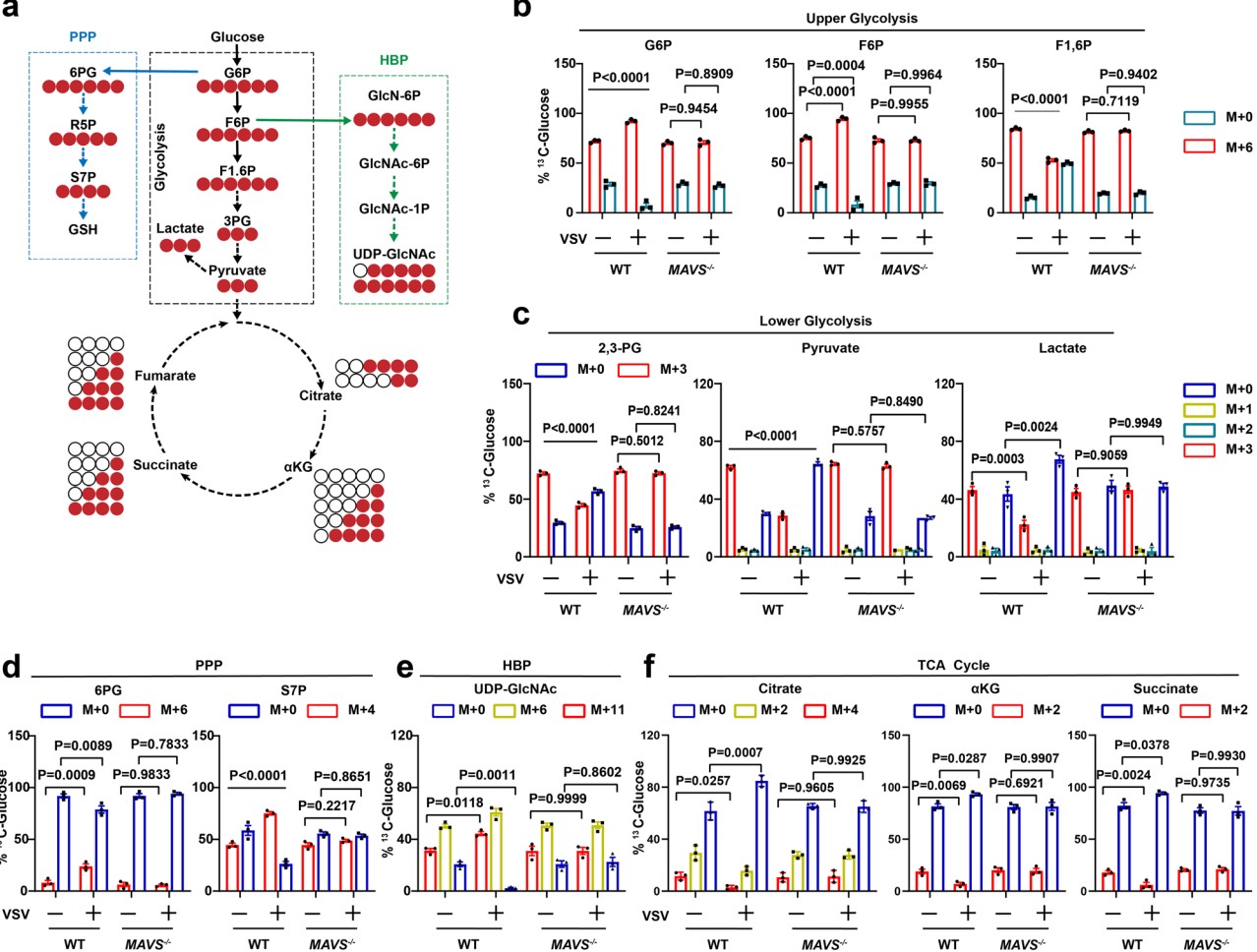

**Fig. 1 | RLR activation shifts glucose flux from glycolysis to PPP and HBP via MAVS. a** Schematic of [13]C[6]-glucose carbon labeling through glycolysis (upper and lower glycolysis), HBP, PPP, and TCA cycle. **b**–**f** WT and *Mavs*[−/−] BMDMs were infected with or without VSV (MOI = 1) for 6 h. [13]C-glucose incorporation into upper glycolytic metabolites (**b**), lower glycolysis (**c**), PPP (**d**), HBP (**e**), and the TCA cycle (**f**) were analyzed (*n* = 3 mice per condition, two-way ANOVA, mean ± SEM). See also Supplementary Fig. 1, 2. Source data are provided as a Source Data file.

higher $^{13}C_6$-glucose incorporation into G6P, F6P, 6-phosphogluconate (6PG), sedoheptulose-7-phosphate (S7P), and UDP-GlcNAc, with lower $^{13}C_6$-glucose incorporation into fructose-1,6-bisphosphate (F1,6 P), 2,3-phosphogluconate, pyruvate, lactate, citrate, α-ketoglutaric, and succinate (Fig. 1b–f, and Supplementary Data 1). These findings suggest that VSV infection increases glucose flux to the PPP (from G6P to S7P) and the HBP (from F6P to UDP-GlcNAc). However, VSV infection did not affect the intermediate metabolites involved in glucose metabolism in $Mavs^{-/-}$ BMDMs (Fig. 1b–f, and Supplementary Data 1).

We measured metabolites that can be synthesized using 1,2-$^{13}C$-glucose incorporation through glycolysis, the PPP, and the TCA cycle (Supplementary Fig. 1g). Consistent with data obtained from $^{13}C_6$-glucose tracing metabolomics, we found that VSV infection resulted in increased glucose flux to the PPP (from G6P to ATP), accompanied by lower 1,2-$^{13}C$-glucose incorporation into glycolysis and the TCA cycle (from F1,6 P to lactate) (Supplementary Fig. 1h–j, and Supplementary Data 3). However, VSV infection did not affect 1,2-$^{13}C$-glucose incorporation in $Mavs^{-/-}$ BMDMs (Supplementary Fig. 1h–j, and Supplementary Data 3). The amide nitrogen donated by glutamine is transmitted to downstream amino sugar metabolites, including N-acetylneuraminic acid (Neu5Ac) and N-acetylmannosamine (ManNAc) (Supplementary Fig. 2a). To further evaluate HBP flux, we performed a transfer of $^{15}N$ from [amide-$^{15}N$] glutamine ([γ-$^{15}N$] glutamine) to UDP-HexNAc. As shown in Supplementary Fig. 2b–f and Supplementary Data 4, VSV infection result in increased HBP intermediates, whereas MAVS ablation abolishes the effect of VSV on HBP intermediates. In line with previous results, a metabolomic analysis indicated that poly(I:C) treatment shifted glucose metabolism from glycolysis to PPP and the HBP, whereas MAVS knockdown removed the effect of poly(I:C) on glucose metabolism reprogramming (Supplementary Fig. 2g–j, and Supplementary Data 4).

We next asked whether MAVS controls the activity and expression levels of critical markers of glucose metabolism in response to RLR activation. We found that VSV infection increased mRNA levels of $Glut1$ and $Glut4$ in WT BMDMs, but the induction was impaired in $Mavs^{-/-}$ BMDMs (Fig. 2a). Similar results were obtained in poly(I:C)-treated THP-1 cells (Supplementary Fig. 3a). Consistently, MAVS overexpression induced HK activity, whereas MAVS knockdown reduced HK activity in HepG2 and THP-1 cells (Fig. 2b and Supplementary Fig. 3b). Pyruvate, lactate, and succinate assay kits were used for the TCA cycle intermediate levels in BMDMs; we found that pyruvate, lactate, and succinate levels were depressed in response to VSV infection, while there were no changes in MAVS ablated cells (Fig. 2c–e). We also observed that poly(I:C) stimulation attenuated pyruvate, lactate, and succinate levels, which were abolished by sh-MAVS (Supplementary Fig. 3c–e). Next, we sought to determine the role of MAVS on G6PD activity and the PPP pathway. As shown in Fig. 2f, g, VSV infection enhanced G6PD activity and G6PD dimer formation in WT BMDMs but not $Mavs^{-/-}$ BMDMs. Consistent with G6PD activity, further analysis revealed that cellular NADPH levels increased during VSV infection, accompanied by a reduction in NADP$^+$/NADPH ratios in WT BMDMs, but not in $Mavs^{-/-}$ BMDMs (Fig. 2h, i). Similarly, MAVS knockdown abolished poly(I:C)-induced G6PD activity and NADPH levels, accompanied by an induction in NADP$^+$/NADPH ratios (Supplementary Fig. 3f–h). Two GFPT isoforms (GFPT1 and GFPT2) are critical enzymes in the HBP pathway[20]. VSV infection increased mRNA levels of $Gfpt2$, but not GFPT1, whereas MAVS ablation abolished the effect of VSV on $Gfpt2$ expression (Fig. 2j). VSV, SeV, and IAV also increased GFPT2 mRNA levels in a time-dependent manner in A549 cells and HeLa cells (Supplementary Fig. 3i–k). Knockdown of MAVS inhibits poly(I:C)-induced GFPT2 expression, whereas poly(I:C) and MAVS did not affect GFPT1 expression (Supplementary Fig. 3l). Interesting, GFPT activity elevation was detected as early as 1 h after VSV infection, suggesting that VSV induce GFPT activity earlier than GFPT2 expression (Supplementary Fig. 3m). Knockdown of MAVS inhibits

VSV-induced GFPT activity (Supplementary Fig. 3m). As expected, VSV-induced HBP end-product UDP-GlcNAc levels and total protein O-GlcNAcylation in WT BMDMs but not $Mavs^{-/-}$ BMDMs (Fig. 2k and Supplementary Fig. 3n). MAVS knockdown reduced poly(I:C)-induced UDP-GlcNAc levels (Supplementary Fig. 3o). These findings suggest that RLR activation shifts energy metabolism from glycolysis to PPP and the HBP via MAVS.

## MAVS subcellular localization has distinct functions on glucose metabolism reprogramming

To locate the subcellular localization of MAVS during glucose metabolism, we constructed a plasmid by replacing the MAVS localization motif with a set of domains (Supplementary Fig. 4a). To determine whether each MAVS allele would direct the protein to a single compartment, $Mavs^{-/-}$ BMDMs were transfected with each MAVS allele, and cell fractionation and Western blot experiments were performed (Supplementary Fig. 4b, c). As shown in Supplementary Fig. 4c, full-length MAVS (MAVS-WT) was located in the cytosol (Cyto), mitochondria (Mito), peroxisomes (Pex), and MAMs. MAVS-Mito, MAVS-Pex, and MAVS-MAMs were found in mitochondria, peroxisomes, and MAMs, respectively (Supplementary Fig. 4c). We next explored whether MAVS subcellular localization would regulate glucose metabolism reprogramming and found that MAVS alleles, except MAVS-Cyto, increased mRNA levels of $Glut1$ and $Glut4$ (Supplementary Fig. 4d). MAVS only in mitochondria (MAVS-WT and -Mito) increased HK activity (Supplementary Fig. 4e). More importantly, MAVS-Mito enhanced glycolysis and the TCA cycle but did not affect the HBP or the PPP (Fig. 3a–e). MAVS-Pex reduced the incorporation of glucose-derived carbon into glycolysis and the TCA cycle but had no impact on the HBP (Fig. 3a–e, and Supplementary Data 2). Indeed, MAVS-Pex increased glucose flux to the PPP, suggesting that MAVS in peroxisomes increase glucose flux to the PPP starting from G6P (Fig. 3a–e). MAMs-located MAVS upregulated incorporation of glucose-derived carbon into the HBP but downregulated incorporation of glucose-derived carbon into F6P, F1,6P, 2,3-phosphogluconate, pyruvate, lactate, citrate, α-ketoglutarate, and succinate (Fig. 3a–e). However, MAMs-located MAVS did not affect the incorporation of glucose-derived carbon into G6P, 6-phosphogluconate, or S7P, suggesting that MAMs-located MAVS increases glucose flux to the HBP starting from F6P (Fig. 3a–e). MAVS only on peroxisomes (MAVS-WT and -Pex) increased NADPH levels and reduced NADP$^+$/NADPH ratios (Supplementary Fig. 4f, g). We also performed tracing experiments using 1,2-$^{13}C$-glucose. MAVS-Mito increased 1,2-$^{13}C$-glucose incorporation into glycolysis and the TCA cycle but not the PPP (Supplementary Fig. 5a–i, and Supplementary Data 5). Conversely, MAVS-Pex reduced 1,2-$^{13}C$-glucose incorporation into glycolysis and the TCA cycle but enhanced 1,2-$^{13}C$-glucose incorporation into the PPP (Supplementary Fig. 5a–i, and Supplementary Data 5). We focused on the HBP using [γ-$^{15}N$] glutamine. As shown in Supplementary Fig. 6a and Supplementary Data 6, MAVS only in MAMs (MAVS-WT and -MAMs) increased [γ-$^{15}N$] glutamine incorporation into the HBP. Similar results were obtained using metabolomic analysis (Supplementary Fig. 6b, and Supplementary Data 6). These findings suggest that peroxisome-located MAVS directs glucose flux to the PPP, while MAMs-located MAVS directs glucose flux to the HBP.

## G6PD is critical for type III IFN production, and GFPT2 is critical for type I IFN production

MAVS activates downstream transcription factors upon RLR activation and induces type I IFN, type III IFN, and inflammatory cytokine expression[21,22]. To test whether the PPP and the HBP would regulate MAVS-mediated signaling, we used pharmacologic methods, including G6PDi-1 (G6PDi) (inhibitor of G6PD), 6-aminonicotinamide (6-AN) (inhibitor of the oxidative branch), azaserine (Aza) (inhibitor of GFPT), and OSMI-1 (inhibitor of O-GlcNAc transferase, OGT) (Fig. 4a). PPP

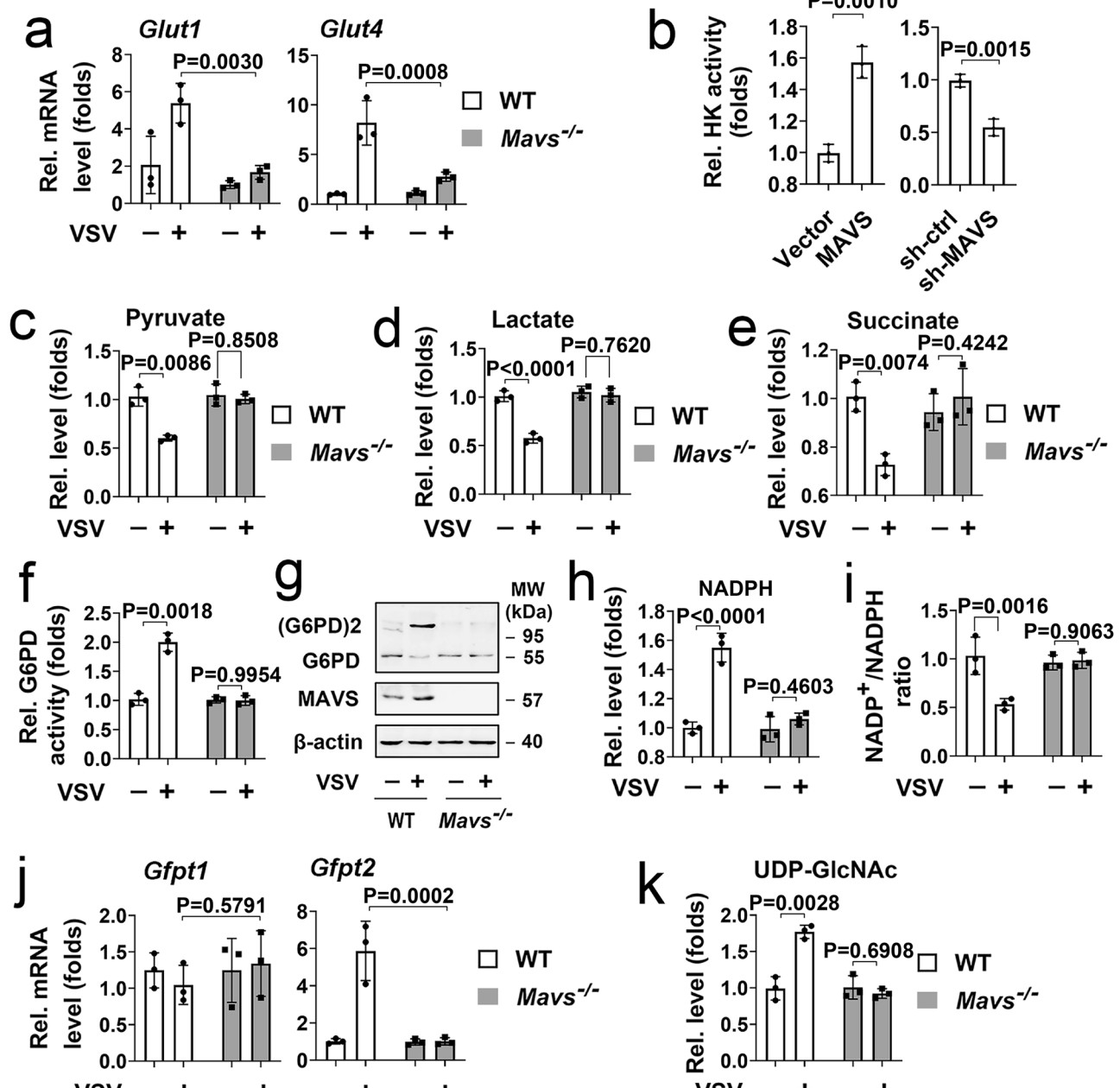

**Fig. 2 | RLR activation alters intermediates of glucose metabolism via MAVS.**
**a** WT and *Mavs*⁻/⁻ BMDMs were infected with or without VSV (MOI = 1) for 6 h before qPCR analyses (*n* = 3 mice per condition, repeated measures two-way ANOVA). **b** HepG2 cells were transfected with control vector or pCMV-MAVS (left panel), shRNA-control, or shRNA-MAVS (right panel) for 36 h, followed by an analysis of mitochondria HK activity (Data represent the means ± SD, two-sided Student's *t*-test). **c**–**i** WT and *Mavs*⁻/⁻ BMDMs were infected with or without VSV (MOI = 1) for 6 h,

followed by measuring total pyruvate (**c**), lactate (**d**), and succinate (**e**) levels, G6PD activity (**f**), or G6PD dimerization (**g**), and NADPH (**h**) and NADP⁺/NADPH (**i**) ratio levels. **j** WT and *Mavs*⁻/⁻ BMDMs were infected with or without VSV (MOI = 1) for 6 h before qPCR analyses. **k** WT and *Mavs*⁻/⁻ BMDMs were infected with or without VSV (MOI = 1) for 6 h, followed by measuring UDP-GlcNAc levels. Data in (**c**–**f**) and (**h**–**k**) are presented as means ± SEMs, *n* = 3 per condition, two-way ANOVA. See also Supplementary Fig. 3. Source data are provided as a Source Data file.

inhibition with G6PDi or 6-AN suppressed VSV-induced expression of cytokines and IFN-λ1 but not IFN-β (Fig. 4b and Supplementary Fig. 7a). Inhibiting GFPT or OGT resulted in low mRNA levels of cytokines and IFN-β but not IFN-λ1 in response to a VSV challenge (Fig. 4c and Supplementary Fig. 7b). Those findings suggest that the PPP controls type III IFN production, and the HBP controls type I IFN production. To understand how PPP and HBP regulates IFN production during RLR activation, we examined the phosphorylation of TBK1 and IRF3. Consistent with decreased IFN expression, the levels of TBK1 and IRF3 phosphorylation robustly declined in 6-AN or OSMI-1 treated cells in response to poly(I:C) stimulation (Supplementary

Fig. 7c). Previous studies showed that MAVS subcellular localizations determine the class of IFN synthesis[21,22]. We examined whether the PPP and the HBP participate in this process. As shown in Fig. 4d–h, G6PDi inhibited MAVS-WT and MAVS-Pex (but not MAVS-Mito and MAVS-MAMs) regulated the expression of proinflammatory cytokines and IFN-λ1. Aza inhibited MAVS-WT and MAVS-MAMs (but not MAVS-Mito and MAVS-Pex) regulated the expression of proinflammatory cytokines and IFN-β (Fig. 4i–m). The effect of the PPP and the HBP on MAVS allele-regulated VSV replication was evaluated. As shown in Supplementary Fig. 7d, e, inhibiting G6PD abolished peroxisomal MAVS (MAVS-WT and -Pex)-regulated VSV replication,

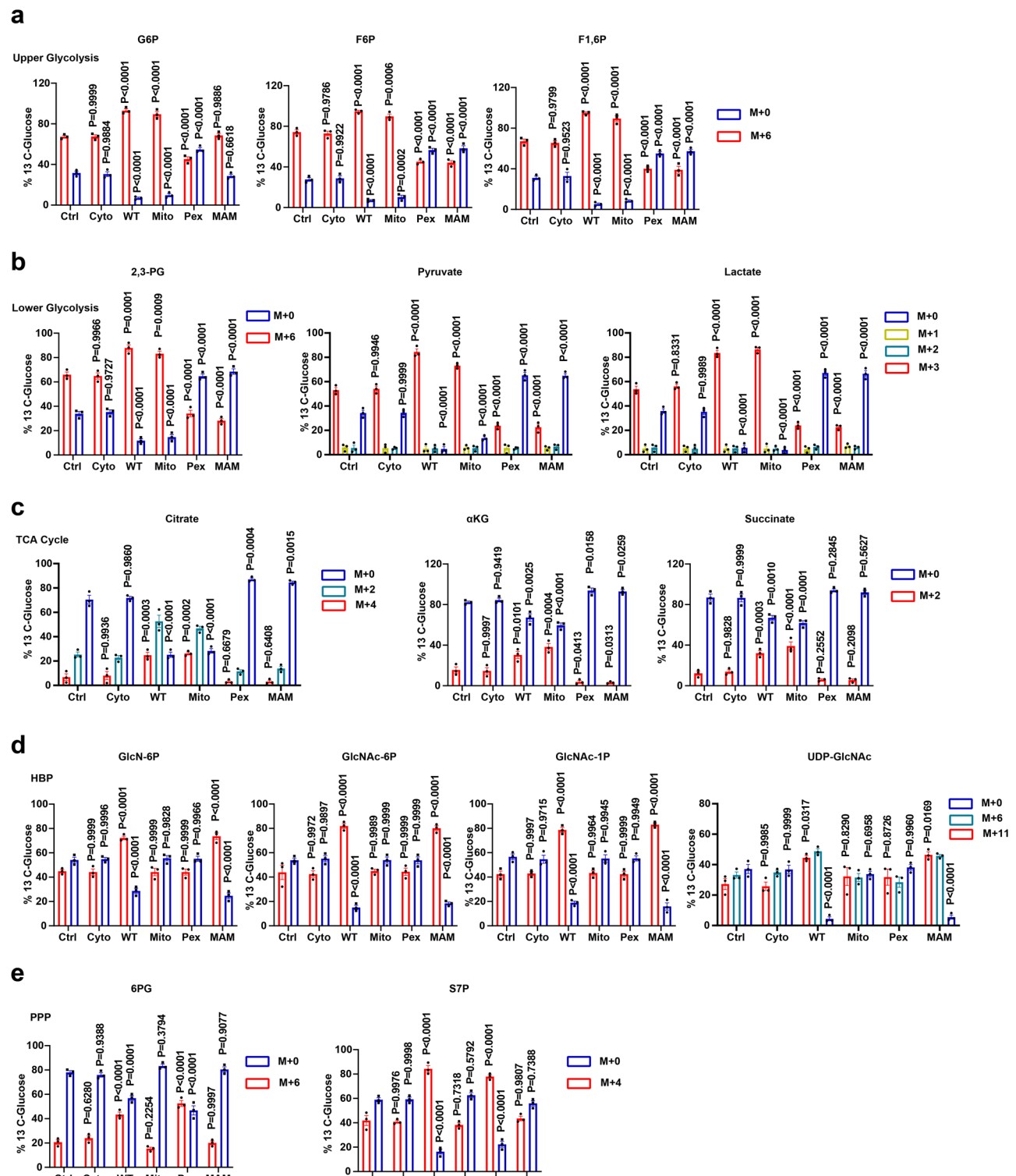

**Fig. 3 | MAVS subcellular localization is critical for glucose metabolism reprogramming. a**–**e** *Mavs*[−/−] BMDMs were transfected with a control vector or indicated MAVS alleles for 24 h. $^{13}C_6$-glucose incorporation into upper glycolytic metabolites (**a**), lower glycolysis (**b**), the TCA cycle (**c**), the HBP (**d**), and the PPP (**e**) are presented as means ± SEMs, *n* = 3 per condition, repeated measures one-way ANOVA. See also Supplementary Figs. 4–6. Source data are provided as a Source Data file.

whereas inhibiting GFPT abolished MAMs-located MAVS (MAVS-WT and -MAMs)-regulated VSV replication.

To further determine the role of G6PD and GFPT2 in RLR signaling, we designed two specific short hairpin RNAs (shRNAs) for G6PD (shRNA-G6PD #1 and #2) and GFPT2 (shRNA-GFPT2 #1 and #2) and tested their efficiency (Supplementary Fig. 8a, b). ShRNA-G6PD#1 and shRNA-GFPT2#1 was selected for the experiments described below.

We observed that G6PD knockdown suppressed poly(I:C)-induced mRNA levels of proinflammatory cytokines and IFN-λ1 but not IFN-β (Supplementary Fig. 8c). However, GFPT2 knockdown suppressed poly(I:C)-induced mRNA levels of proinflammatory cytokines and IFN-β but not IFN-λ1 (Supplementary Fig. 8d). Further studies showed that G6PD knockdown inhibited MAVS-WT and MAVS-Pex induced the expression of proinflammatory cytokines and IFN-λ1 but not IFN-β

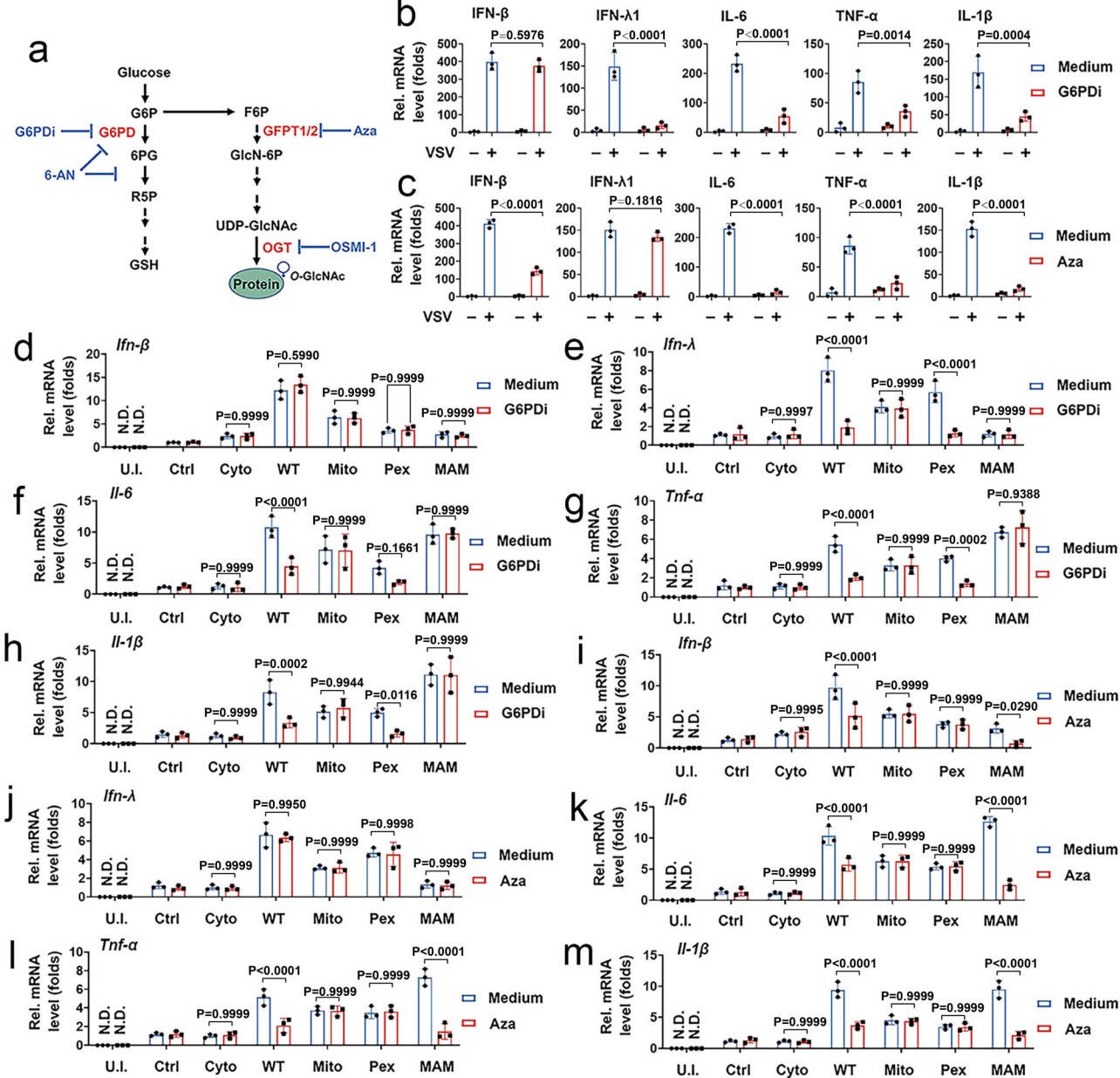

**Fig. 4 | The PPP and the HBP regulate antiviral immune responses in response to RLR activation. a** Schematic of PPP and HBP. The enzyme/pathway targeted by each inhibitor is shown. **b** THP-1 cells were infected with or without VSV (MOI = 1) for 6 h and treated with or without G6DPi (50 μM for 4 h) before qPCR analyses. **c** Experiments were performed as described in (**b**), except that Aza (0.5 mM for 6 h) were used. **d–h** *Mavs⁻/⁻* BMDMs were transfected with a control vector or indicated MAVS alleles for 24 h and treated with or without G6DPi (50 μM for 4 h) before qPCR. **i–m** Experiments were performed as described in (**d–h**), except that Aza (0.5 mM for 6 h) was used. Data in (**b**) and (**c**) are presented as means ± SD, two-way ANOVA. Data in (**d–m**) are expressed as means ± SEMs, *n* = 3 mice per condition, two-way ANOVA. See also Supplementary Figs. 7–9. Source data are provided as a Source Data file.

(Supplementary Fig. 8e–i). By contrast, GFPT2 knockdown inhibited MAVS-WT- and MAVS-MAMs-induced expression of proinflammatory cytokines and IFN-β but not IFN-λ1 (Supplementary Fig. 8j–n). To test whether the PPP and the HBP would regulate VSV replication via different classes of IFN, we generated type I IFN receptor-deficient A549 cells (IFNAR1⁻/⁻ cells, AR⁻/⁻ cells) and type III IFN receptor-deficient A549 cells (IFNLR1⁻/⁻ cells, LR⁻/⁻ cells) using CRISPR/Cas9 technology and confirmed the absence of IFNAR1 and IFNLR1 (Supplementary Fig. 9a, b). As expected, overexpression of G6PD inhibited VSV replication accompanied by higher ISG56 expression in WT and AR⁻/⁻ cells but not in LR⁻/⁻ cells (Supplementary Fig. 9c). By contrast, overexpression of GFPT2 inhibited VSV replication accompanied by higher ISG56 expression in WT and LR1⁻/⁻ cells but not in AR1⁻/⁻ cells

(Supplementary Fig. 9d). Conversely, G6PD knockdown induced VSV replication accompanied by lower ISG56 expression in WT and AR⁻/⁻ cells but not in LR⁻/⁻ cells (Supplementary Fig. 9e). By contrast, GFPT2 knockdown induced VSV replication accompanied by lower ISG56 expression in WT and LR1⁻/⁻ cells but not in AR1⁻/⁻ cells (Supplementary Fig. 9f). Similar results were obtained by using anti-IFNα/β and anti-IFNλ neutralizing antibodies (Supplementary Fig. 9g, h). Overexpression of G6PD and MAVS-WT or -Pex synergistically inhibited VSV replication accompanied by higher *Isg56* expression (Supplementary Fig. 9i). However, anti-IFNλ (but not anti-IFNα/β) neutralizing antibodies diminished MAVS-Pex-regulated VSV replication and *Isg56* expression (Supplementary Fig. 9i). By contrast, overexpression of GFPT2 and MAVS-WT or -MAMs synergistically inhibited VSV

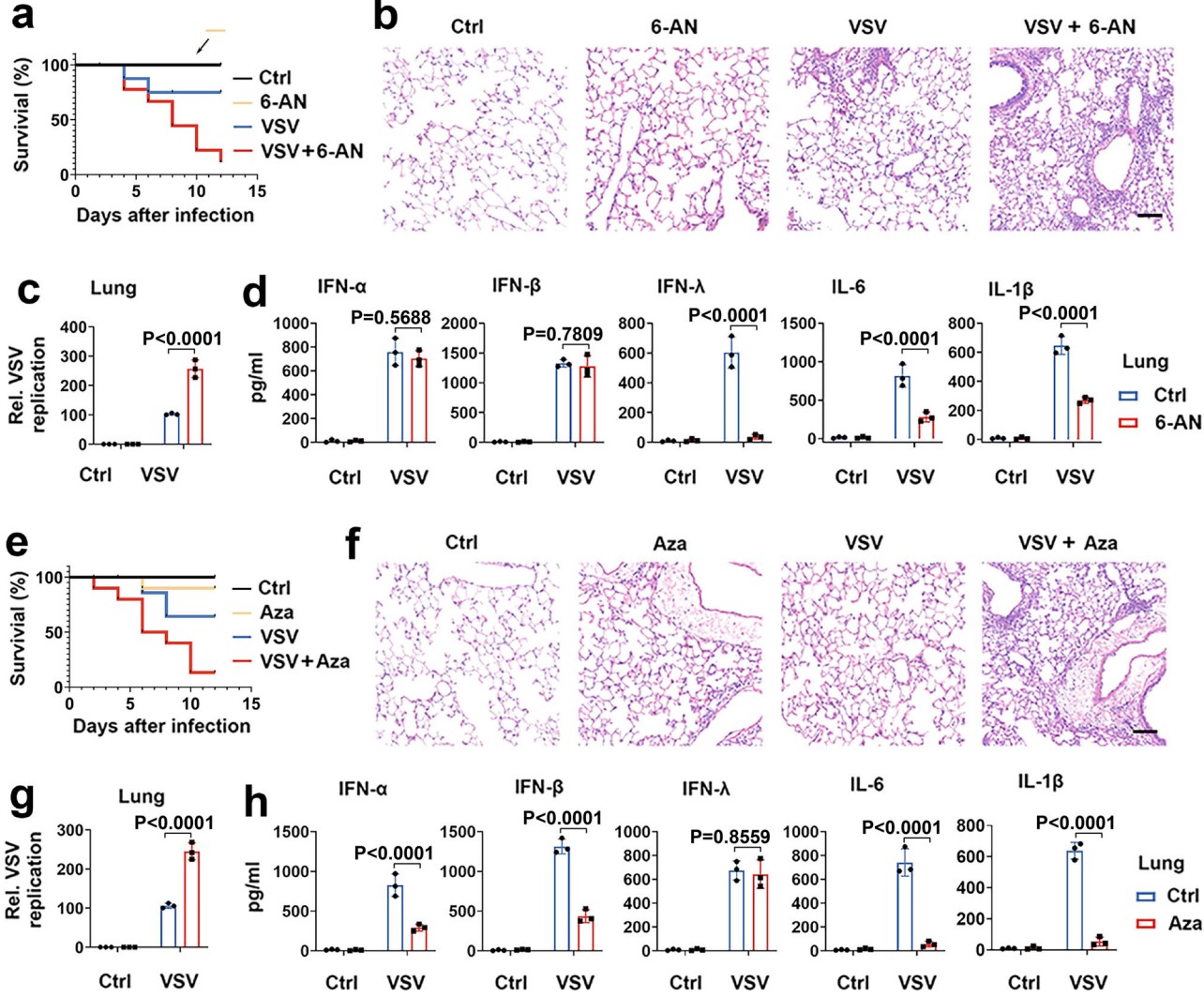

**Fig. 5 | The PPP and the HBP are critical for the activation of antiviral immune signaling in vivo. a** C57BL/6 mice were infected with VSV ($2 \times 10^7$ pfu/g) and treated with PBS (Ctrl) or 6-AN (4 mg/kg per day) by intraperitoneal injection. Survival curves show data collected until day 12 after infection. Statistical analysis was performed using the log-rank test ($n = 5$ for each group). **b, c** C57BL/6 mice were treated with PBS or 6-AN (4 mg/kg per day) for 24 h and infected with VSV ($2 \times 10^7$ pfu/g) for 24 h, followed by an analysis of the lung tissue (**b**), VSV RNA in the lungs (**c**). Scale bar, 100 μm. **d** C57BL/6 mice were treated with PBS or 6-AN (4 mg/kg per day) for 24 h and infected with VSV ($2 \times 10^7$ pfu/g) for 24 h, followed by an analysis of levels of proinflammatory cytokines and IFN in the lung. **e**–**h** Experiments were performed as described in (**a**–**d**), except that azaserine (Aza) (2.5 mg/kg per day) was used. Data in (**c**), (**d**), (**g**) and (**h**) are presented as means ± SEMs, $n = 3$ mice per condition, two-way ANOVA. See also Supplementary Fig. 10. Source data are provided as a Source Data file.

replication accompanied by higher *Isg56* expression (Supplementary Fig. 9j). Anti-IFNα/β (but not anti-IFNλ) neutralizing antibodies diminished MAVS-MAMs-regulated VSV replication and *Isg56* expression (Supplementary Fig. 9j). These findings suggest that G6PD was involved in PPP-regulated MAVS-Pex-induced type III IFN and cytokine expression, whereas GFPT2 was involved in HBP-regulated MAVS-MAMs induced type I IFN and cytokine expression.

### The PPP and the HBP regulate antiviral innate immune responses through different classes of IFN in vivo

We tested the role of PPP in antiviral innate immune responses in vivo. As shown in Fig. 5a, the 6-AN treatment produced significantly greater VSV-induced mortality than control mice. Histological analysis of the lung tissue showed more significant infiltration of immune cells and injury in 6-AN-treated mice than in control mice (Fig. 5b). 6-AN-treated mice demonstrated significantly higher levels of VSV replication in the spleen, liver, and lungs, suggesting insufficient control of viral replication (Fig. 5c and Supplementary Fig. 10a). As expected, enzyme-linked immunosorbent assays indicated that 6-AN treatment inhibited the

protein levels of cytokines and IFN-λ (but not IFN-β) in the spleen, liver, and lungs after VSV challenge (Fig. 5d and Supplementary Fig. 10b, c). We next explored the role of the HBP on antiviral innate immune responses in vivo. As shown in Fig. 5e, f, Aza treatment was significantly susceptible to VSV-induced lethality, accompanied by significantly more severe lung injury. Aza-treated mice demonstrated significantly higher levels of VSV replication in the spleen, liver, and lungs, accompanied by lower protein levels of IFN-α, IFN-β, and cytokines (but not IFN-λ) in the spleen, liver, and lungs (Fig. 5g, h and Supplementary Fig. 10d–f). These findings suggest that the PPP regulates type III IFN production, and the HBP regulates type I IFN production in RLR signaling in vivo.

### GFPT2 interacts with a ternary complex

Because Aza and sh-GFPT2 inhibit MAVS-regulated signaling, we hypothesized that GFPT2 would directly interact with the MAVS complex. Coimmunoprecipitation (Co-IP) results indicated an association of GFPT2 with MAVS, TRAF2, and TRAF6, but no association with the other examined proteins, including RIG-I, TAK1, IκB kinase β, TRAF3, NEMO (Fig. 6a, b and Supplementary Fig. 11a). We performed

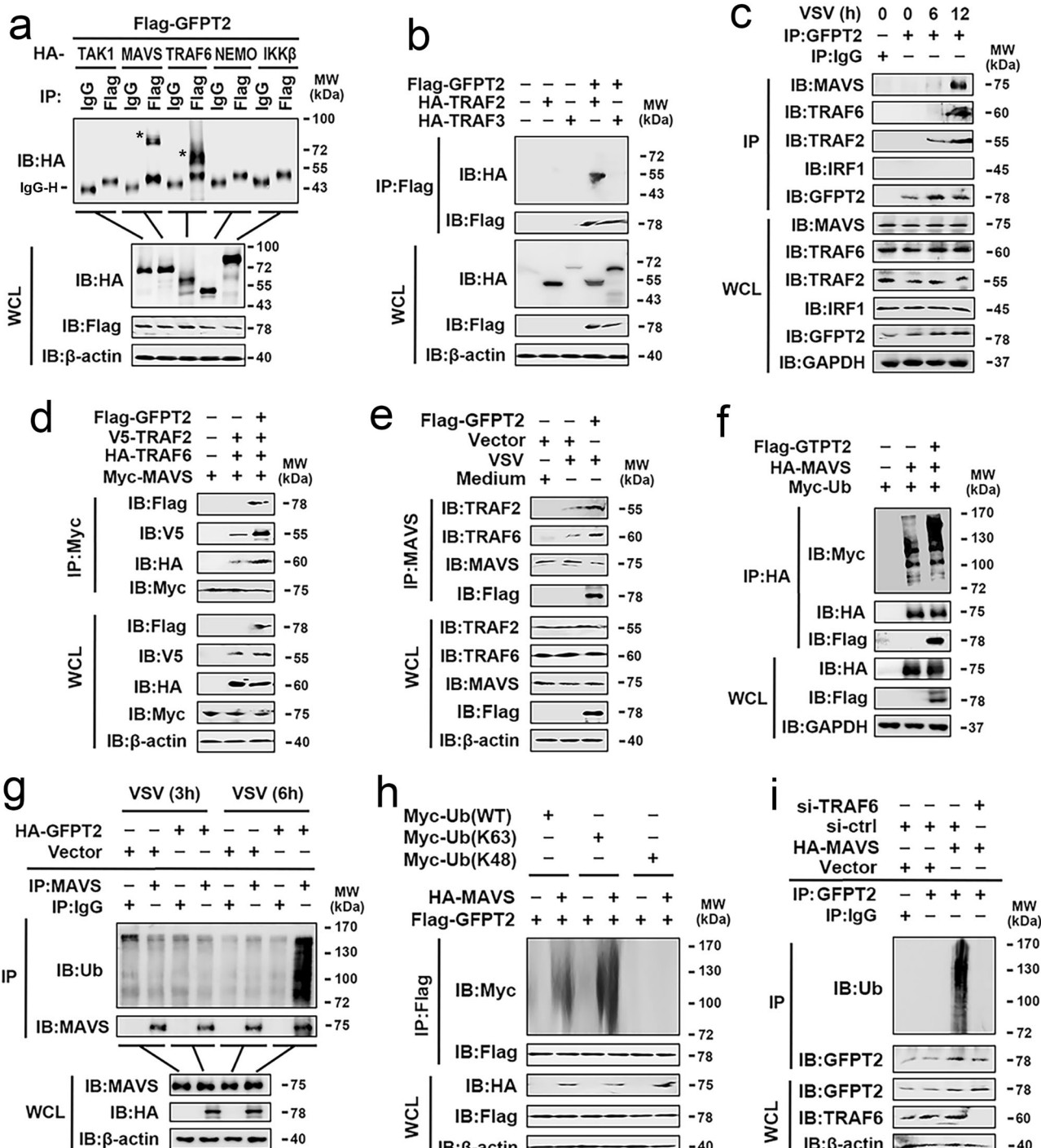

**Fig. 6 | GFPT2 associates with the MAVS/TRAF2/TRAF6 complex. a, b** HEK293 cells were transfected with indicated plasmids for 48 h. Co-IP and immunoblot analyses were performed using the indicated antibodies. **c** A549 cells were mock-infected or infected with VSV (MOI = 1) for the indicated times and subjected to Co-IP and immunoblotting analysis with the indicated antibodies. **d** HEK293 cells were transfected with indicated plasmids for 48 h. Co-IP and immunoblot analyses were performed with the indicated antibodies. **e** A549 cells were transfected with vector control or Flag-GFPT2 for 36 h and infected with VSV (MOI = 1) for 12 h. Co-IP and immunoblot analyses were performed with the indicated antibodies. **f** HEK293 cells were transfected with HA-MAVS, FLAG-GFPT2, and myc-tagged ubiquitin (Ub) plasmids for 24 h. Co-IP and immunoblot analyses were performed with the indicated antibodies. **g** HEK293 cells were transfected with vector or GFPT2 expression plasmid. Twenty-four hours later, the cells were infected with VSV (MOI = 1) for 3 h or 6 h, respectively. Co-IP and immunoblot analyses were performed with the indicated antibodies. **h** A549 cells were transfected with indicated plasmids for 48 h. Co-IP and immunoblot analyses were performed with the indicated antibodies. **i** A549 cells were transfected with indicated plasmids or siRNAs for 36 h. Co-IP and immunoblot analyses were performed with the indicated antibodies. All experiments were repeated at least three times. See also Supplementary Fig. 11, 12. Source data are provided as a Source Data file.

endogenous Co-IP experiments and found that VSV stimulation induced GFPT2 associated with MAVS, TRAF2, and TRAF6 (but not IRF1) (Fig. 6c). To map the region of MAVS, TRAF6, and TRAF2 that interact with GFPT2, we constructed truncation mutants of MAVS, TRAF6, and TRAF2 (Supplementary Fig. 11b–d, upper panel). We found that the CARD domain of MAVS (amino acids 1-181), CC domains of TRAF6 (amino acids 288-357), and the TRAF-C domain of TRAF2 (amino acids 355-501) were required for its interaction with GFPT2 (Supplementary Fig. 11b–d, lower panel). To map the region of GFPT2 that interacts with MAVS, TRAF6, and TRAF2, we constructed truncation mutants of GFPT2 (Supplementary Fig. 11e, upper panel). Co-IP experiments showed that MAVS interacted with all truncations of GFPT2, TRAF6 interacted with the D1 and D3 domains of GFPT2 (amino acids 1−360 and 531-682), and TRAF2 interacted with the D3 domain of GFPT2 (amino acids 531-682) (Supplementary Fig. 11e−g). Next, we assessed the relationship between GFPT2, MAVS, TRAF2, and TRAF6 in response to RLR stimulation. Co-IP and endogenous Co-IP experiments indicated that GFPT2 induced the formation of a ternary complex composed of MAVS, TRAF2, and TRAF6 (Fig. 6d, e). Because the ubiquitination of MAVS is a critical modification that promotes downstream signaling[23], we determined whether GFPT2 would affect the polyubiquitination of MAVS. As shown in Fig. 6f, GFPT2 promoted polyubiquitination of MAVS. Endogenous Co-IP experiments revealed that GFPT2 overexpression increased VSV-induced polyubiquitination of MAVS (Fig. 6g). The ubiquitination of GFPT2 is essential for its activity[24]. We next examined the role of MAVS on GFPT2 ubiquitination. As shown in Fig. 6h, MAVS enhanced K63-linked (but not K48-linked) polyubiquitination of GFPT2. Endogenous Co-IP experiments indicated that MAVS enhanced polyubiquitination of GFPT2 via TRAF6 (Fig. 6i).

We next explored the role of GFPT2 on the MAVS-mediated downstream signaling pathway. As shown in Supplementary Fig. 12a, b, GFPT2 overexpression increased the phosphorylation of IκBα, IKKα/β, and IRF3 and GFPT2 knockdown decreased these phosphorylation events. Previous studies indicated that transcription factors IRF3 and NF-κB are required for the induction of type I IFN, and IRF1 and NF-κB are required for the induction of type III IFN[21,22]. Using luciferase activity reporter assays, we showed that GFPT2 knockdown inhibited VSV- and MAVS-mediated activation of NF-κB and IFN-stimulated response element (ISRE) (that typically reports IRF3 activity) (Supplementary Fig. 12c, d). As expected, GFPT2 knockdown failed to affect the activation of IRF1 in response to VSV infection and MAVS transfection (Supplementary Fig. 12e). These findings suggest that RLR activation induces the interaction of GFPT2 with the MAVS/TRAF2/TRAF6 ternary complex, thereby ubiquitinating MAVS and GFPT2 and leading to the activation of the HBP and type I IFN.

## G6PD interacts with a binary complex
Next, we determined whether G6PD would interact with the MAVS complex. Co-IP experiments indicated that G6PD interacts with MAVS and TRAF6 but not TRAF2 and IRF1 (Fig. 7a). Endogenous Co-IP experiments revealed that G6PD interacts with MAVS, TRAF6, and IRF1 (but not TRAF2 or TRAF3) after stimulation with VSV (Fig. 7b). These findings suggest that G6PD interacts with MAVS and TRAF6, leading to MAVS complex recruiting IRF1, although G6PD did not directly coprecipitate with IRF1. Using MAVS and TRAF6 truncation mutants, we found that the CARD domain of MAVS (amino acids 1-181) and the TRAF-C domain of TRAF6 (amino acids 357-522) were required for its interaction with G6PD (Supplementary Fig. 13a, b). To map the region of G6PD that interacts with MAVS and TRAF6, we constructed truncation mutants of G6PD (Supplementary Fig. 13c, upper panel). Co-IP experiments showed that MAVS interacts with interacted with the C-terminal domain of G6PD (amino acids 211-515), and TRAF6 interacted with the N-terminal domain of G6PD (amino acids 1-210) (Supplementary Fig. 13c, d). As expected, Co-IP and endogenous Co-IP

experiments revealed that G6PD induced the formation of a binary complex composed of MAVS and TRAF6 (Fig. 7c, d). We next determined whether G6PD would play a role in the ubiquitination of IRF1. In an overexpression system, Flag-G6PD enhanced K63-linked (but not K48-linked) polyubiquitination of MAVS (Fig. 7e). G6PD-enhanced IRF1 polyubiquitination was markedly attenuated by TRAF6 knockdown (Fig. 7f). TRAF6 or MAVS knockdown inhibited VSV-induced G6PD dimer formation (Fig. 7g). Luciferase activity reporter assays revealed that G6PD knockdown weakly inhibited VSV- and MAVS-mediated activation of NF-κB and ISRE but significantly inhibited VSV- and MAVS-mediated activation of IRF1 (Supplementary Fig. 13e, f). These findings suggest that RLR activation induces the G6PD interaction with MAVS and TRAF6, leading to IRF1 polyubiquitination and G6PD dimer formation, resulting in PPP initiation and IRF1 activation.

## GFPT2 interacts with MAVS/TRAF2/TRAF6 on MAMS, and G6PD interacts with MAVS/TRAF6 on peroxisomes
Because the MAVS complex binds GFPT2 or G6PD to regulate the HBP or the PPP, we suspected these protein complexes formed in different subcellular locations. To address the question, $Mavs^{-/-}$ BMDMs were transfected with MAVS alleles, and Co-IP experiments were performed to check their colocalization. As shown in Fig. 8a, MAMs-located MAVS (MAVS-WT and -MAMs) interacted with GFPT2, TRAF2, and TRAF6, and Pex-located MAVS (MAVS-WT and -Pex) interacted with G6PD and TRAF6. Confocal microscopy analysis revealed that MAVS and GFPT2 were present at MAM during VSV infection (Supplementary Fig. 14a, b). As expected, VSV infection forced MAVS and G6PD translocate to Pex (Supplementary Fig. 14c, d). MAMs-located MAVS (MAVS-WT and -MAMs) increased protein levels of GFPT2 but not GFPT1 (Fig. 8b). Consistently, MAVS-WT and -MAMs induced the polyubiquitination of GFPT2 (Fig. 8c). As expected, MAVS-WT and -Pex enhanced G6PD activity and G6PD dimer formation (Fig. 8d, e). These findings led us to investigate whether MAVS alleles would modulate the localization of GFPT2, G6PD, TRAF2, TRAF3, TRAF6, and IRF1. Mitochondria-located MAVS transfection forced TRAF3 and TRAF6 to translocate to Mito, MAVS-WT and -MAMs transfection forced GFPT2, TRAF2, and TRAF6 to translocate to MAMs, and MAVS-WT and -Pex transfection forced G6PD, TRAF6, and IRF1 to translocate to Pex (Fig. 8f). GFPT2 knockdown inhibited the activation of NF-κB and ISRE in MAVS-WT and -MAMs (but not in MAVS-Mito, -Pex and -Cyto) transfected cells (Fig. 8g, h). GFPT2 knockdown did not affect the activation of IRF1 in all MAVS alleles transfected cells (Fig. 8i). G6PD knockdown weakly inhibited the activation of ISRE and significantly inhibited the activation of NF-κB and IRF1 in MAVS-WT and -Pex (but not in MAVS-Mito, -MAMs, and -Cyto) transfected cells (Fig. 8g-i). Similarly, GFPT2 knockdown inhibited the effect of MAVS-WT and -MAMS on the expression of Isg56 and Isg15, while G6PD knockdown inhibited the effect of MAVS-WT and -Pex on the expression of these genes (Fig. 8j, k). These findings support the notion that MAVS interacts with G6PD and TRAF6 on the Pex and exerts its function on the PPP and type III IFN production. Alternatively, MAVS interacts with GFPT2, TRAF2, and TRAF6 on the MAMs, leading to the activation of the HBP and type I IFN.

## TRAF6, TRAF2, and IRF1 control glucose metabolism reprogramming during VSV infection
Because GFPT2 interacts with the MAVS/TRAF2/TRAF6 ternary complex, and G6PD interacts with the MAVS/TRAF6 binary complex, we next examined the role of other MAVS complex signaling regulators on glucose metabolism. We designed two shRNAs for TRAF6, TRAF2, and IRF1 and tested their efficiency (Supplementary Fig. 15a−c). ShRNA-TRAF6#1, shRNA-TRAF2#1, and shRNA-IRF1#1 was selected for experiments described below. Metabolomic analysis revealed that TRAF6 knockdown abolished VSV-regulated metabolite levels (Supplementary Fig. 15d−g, and Supplementary Data 7). This result is

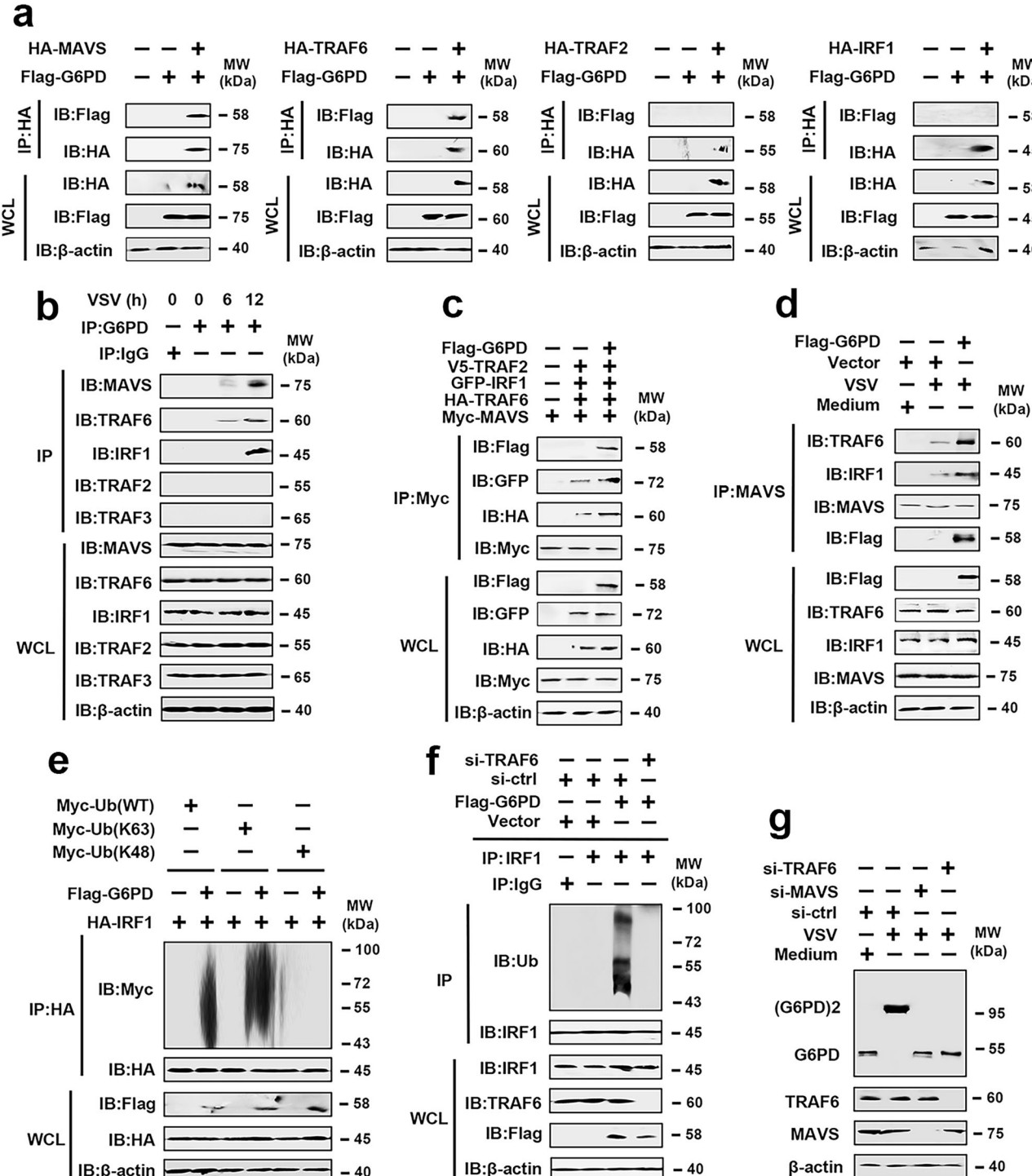

**Fig. 7 | G6PD is associated with the MAVS/TRAF6 complex. a** HEK293 cells were transfected with indicated plasmids for 48 h. Co-IP and immunoblot analyses were performed with the indicated antibodies. **b** THP-1 cells were mock-infected or infected with VSV (MOI = 1) for the indicated times and subjected to Co-IP and immunoblotting analysis with the indicated antibodies. **c** HEK293 cells were transfected with indicated plasmids for 48 h. Co-IP and immunoblot analyses were performed with the indicated antibodies. **d** THP-1 cells were transfected with vector control or Flag-G6PD for 36 h and infected with VSV (MOI = 1) for 6 h. Co-IP and immunoblot analyses were performed with the indicated antibodies. **e** HEK293 cells

were transfected with indicated plasmids for 48 h. Co-IP and immunoblot analyses were performed with the indicated antibodies. **f** THP-1 cells were transfected with vector control, Flag-G6PD, si-ctrl, or si-TRAF6 for 36 h. Co-IP and immunoblot analyses were performed with the indicated antibodies. **g** THP-1 cells were transfected with vector si-ctrl, si-MAVS, or si-TRAF6 for 36 h and infected with VSV (MOI = 1) for 6 h, followed by an analysis of G6PD dimerization. All experiments were repeated at least three times. See also Supplementary Fig. 13. Source data are provided as a Source Data file.

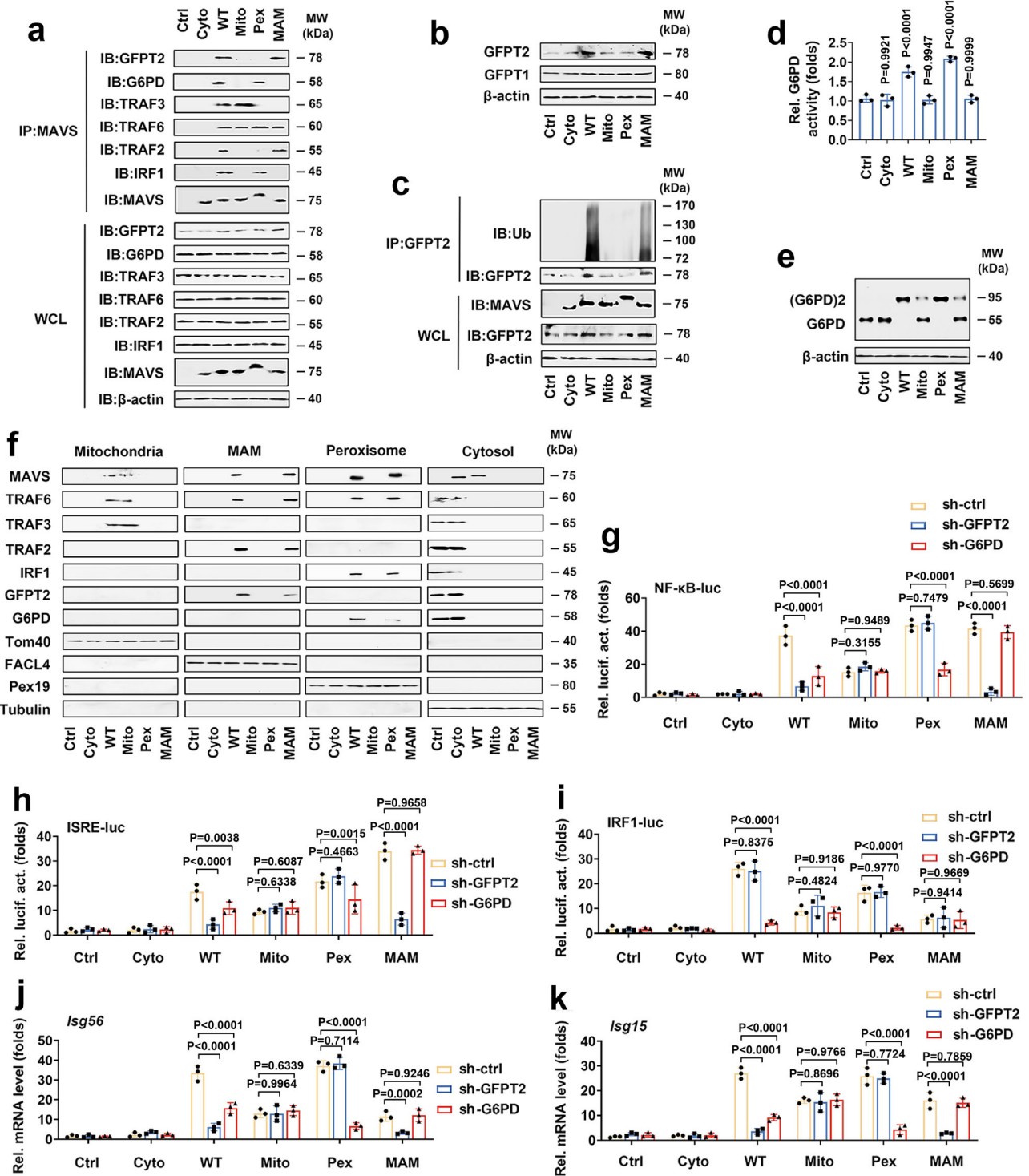

**Fig. 8 | The specific subcellular localization of MAVS regulates different signaling pathways via the recruitment of GFPT2/TRAF6/TRAF2 or G6PD/TRAF6/IRF1. a–c** *Mavs⁻/⁻* BMDMs were transfected with a control vector or indicated MAVS alleles for 48 h. Co-IP and immunoblot analyses (**a, c**) and Western blot analyses (**b**) were performed with the indicated antibodies. **d, e** *Mavs⁻/⁻* BMDMs were transfected with the control vector or indicated MAVS alleles for 48 h, followed by an analysis of G6PD activity (**d**) or G6PD dimerization (**e**) (*n* = 3 mice per condition, means ± SEMs, one-way ANOVA). **f** *Mavs⁻/⁻* BMDMs were transfected with the control vector or indicated MAVS alleles for 48 h. Subcellular fractions were isolated for immunoblot analysis. Fractionation markers: mitochondria (Tom40); MAMs (FACL4); peroxisomes (Pex19); cytosol (Tubulin). **g** *Mavs⁻/⁻* BMDMs were transfected with NF-κB-luc and indicated MAVS alleles for 48 h before luciferase assays. **h, i** Experiments were performed similar to those in (**g**), except ISRE-luc (**h**) or IRF1-luc (**i**) were used. **j, k** *Mavs⁻/⁻* BMDMs were transfected with si-ctrl, si-GFPT2, si-G6PD, or indicated MAVS alleles for 36 h before qPCR analyses. Data in (**a–c, f**) are representative from three independent experiments. Data in (**g–k**) are presented as means ± SEMs, *n* = 3 mice per condition, two-way ANOVA. See also Supplementary Fig. 14. Source data are provided as a Source Data file.

consistent with the previous results showing that TRAF6 interacted with MAVS on mitochondria, peroxisomes, and MAMs (Fig. 8a). TRAF2 and IRF1 knockdown increased the metabolites of glycolysis and the TCA cycle, suggesting that blockade of the PPP and the HBP force glucose metabolism to shift to glycolysis and the TCA cycle (Supplementary Fig. 15d, e, and Supplementary Data 7). Moreover, IRF1 knockdown inhibited VSV-induced levels of PPP metabolites but did not affect HBP metabolites, suggesting that IRF1 increases glucose flux to the PPP (Supplementary Fig. 15f, g). However, TRAF2 knockdown inhibited VSV-induced levels of HBP metabolites but did not affect PPP metabolites, suggesting that TRAF2 increases glucose flux to the HBP (Supplementary Fig. 15f, g, and Supplementary Data 7). Further experiments showed that TRAF6 knockdown (but not TRAF2 and IRF1 knockdown) abolished VSV-regulated HK activity (Supplementary Fig. 15h). High pyruvate and lactate levels were observed in TRAF6, TRAF2, and IRF1 knockdown cells (Supplementary Fig. 15i). We also found that TRAF6 and IRF1 knockdown (but not TRAF2 knockdown) reduced VSV-upregulated G6PD activity and NADPH levels accompanied by induction in NADP$^+$/NADPH ratios (Supplementary Fig. 15j, k). In contrast, TRAF6 and TRAF2 knockdown (but not IRF1 knockdown) inhibited VSV-induced GFPT2 mRNA levels and UDP-GlcNAc levels (Supplementary Fig. 15l, m). However, the GFPT1 mRNA level did not change in TRAF6, TRAF2, and IRF1 knockdown cells (Supplementary Fig. 15l).

We next measured the effect of TRAF6, TRAF2, and IRF1 on MAVS subcellular localization in regulating glucose metabolism reprogramming. As shown in Supplementary Fig. 16a, TRAF6 knockdown inhibited MAVS-WT and -mito-regulated HK activity. TRAF6 knockdown abolished the effect of all MAVS alleles on pyruvate and lactate levels (Supplementary Fig. 16b, c). TRAF2 knockdown abolished MAVS-MAMs-inhibited pyruvate and lactate levels, whereas IRF1 knockdown abolished MAVS-Pex inhibited pyruvate and lactate levels (Supplementary Fig. 16b, c). Consistently, TRAF6 and IRF1 (but not TRAF2) knockdown inhibited peroxisome-located MAVS (MAVS-WT and -Pex), enhancing G6PD activity and cellular NADPH levels accompanied by a reduction in NADP$^+$/NADPH ratios (Supplementary Fig. 16d–f). By contrast, MAMs-located MAVS (MAVS-WT and -MAMs) (but not MAVS-Mto and MAVS-Pex) increased GFPT2 mRNA levels and UDP-GlcNAc levels; TRAF6 and TRAF2 knockdown abolished this effect (Supplementary Fig. 16g, h). These findings suggest that TRAF6 and IRF1 regulate the PPP in peroxisomes, while TRAF6 and TRAF2 regulate the HBP in MAMs.

## Discussion

We identified a previously undescribed mechanism of RLR signaling-regulated glucose metabolism reprogramming, in which RLR stimulation shifts energy metabolism from glycolysis to the PPP and the HBP via MAVS. Upon investigating the mechanisms behind this event, we found that differential MAVS placement alters the types of activated glucose pathways. This diversification is functionally important, as our data indicate that MAVS signaling must occur from distinctive organelles to activate glucose pathways, IFN, and cytokine production.

Studies showed that MAVS is an essential adapter protein mediating innate immunity to RNA viruses[25,26]. However, the role of RLR signaling and MAVS in glucose metabolism remains controversial. Several reports suggested that lactate and succinate are natural suppressors of RLR signaling by targeting MAVS in uninfected cells[17,19]. Poly(I:C) treatment downregulated glucose metabolism, including glycolysis and the TCA cycle, by disassociating MVAS and HK2[19]. Other studies showed that VSV infection enhanced activities of glucose metabolic pathways, and OGT promoted RLR-mediated antiviral immune responses[18]. Further research showed that OGT interacted with MAVS and induced MAVS O-GlcNAcylation on S366[18]. These apparently contradictory effects of RLR signaling on glucose metabolism led us to hypothesize that RLR activation initiates metabolic reprogramming using MAVS as an adapter protein. The central finding of this study (i.e., that MAVS is essential for integrating glucose metabolism and RLR signaling) was established using a complementary set of assays that measured (i) $^{13}C_6$-glucose labeling metabolomics, (ii) 1,2-$^{13}$C-glucose labeling metabolomics, (iii) $^{15}$N-glutamine labeling metabolomics, and (iv) metabolomics. In each of these assays, we found that RLR activation shifts glucose flux from glycolysis to the PPP and the HBP. Further experiments demonstrated that MAVS is a critical site of RLR signaling-regulated metabolism reprogramming.

The localization of MAVS is intriguing. MAVS is located in the mitochondria, peroxisomes, and MAMs[21,22]. MAMs are ER membranes at mitochondria-ER contact sites, where they mediate independent functions[27,28]. On the innate immunity side, peroxisomal MAVS selectively induces type III IFN expression via IRF1, whereas mitochondrial MAVS induces an antiviral response typified by the expression of type I IFN and ISGs. However, to our knowledge, a role for MAMs-localized MAVS in innate immunity has not been established. A landmark study showed that MAVS localized to the mitochondria and interacted with HK2, directing glucose flux into glycolysis in uninfected cells on the glucose metabolism side[17]. However, no study investigated the role of peroxisome- and MAMs-localized MAVS-regulated glucose metabolism. Here, we demonstrated that differential MAVS placement directs the glucose pathways. We propose a model for reprogramming MAVS-dependent IFN signaling and glucose metabolism by considering previous work and our results. In the event of RNA virus infection, partial MAVS dissociated from HK translocated from mitochondria to peroxisomes and MAMs and regulated glucose and innate immunity in two distinct ways. In one, GFPT2 interacts with MAVS, and the MAVS signalosome forms at MAMs by recruiting TRAF6 and TRAF2 into this subcellular location to form a signaling synapse. As a result, glucose flux shifts into the HBP and type I IFN production (Fig. 9a). Conversely, G6PD interacts with MAVS, and the MAVS signalosome forms at peroxisomes by recruiting TRAF6 and IRF1 into this subcellular location to form another signaling synapse. This signaling synapse controls glucose flux shift into the PPP and types III IFN production (Fig. 9a). Inhibiting PPP by G6PDi or 6-AN impairs the interaction between G6PD and MAVS, suppressing type III IFN production. On the other hand, inhibiting HBP by Aza interfere the interaction between GFPT2 and MAVS, reducing type I IFN production (Fig. 9b).

We note that there are some limitations to this study. (i) We focused on glucose metabolic changes regulated by RNA virus infection. In addition to glucose metabolism, substrates derived from two other significant nutrients (i.e., lipids and amino acids) might also be important for MAVS-regulated signaling. (ii) The function of MAVS is controlled at the posttranslational modification level, including ubiquitination, aggregation, and geranyl-geranylation[29,30]. The MAVS signalosome consists of hundreds of proteins[31,32]. Our study only focused on the ubiquitination of MAVS and the TRAF family of the MAVS signalosome. (iii) Patients with deficiencies in the G6PD or GFPT2 were susceptible to recurrent infections and sepsis[33,34]. It remains unknown whether using a series of pharmacological, enhancing G6PD or GFPT2 activity would enhance antiviral response in patients. (iiii) Interesting, a recent study showed that DC increased their glycolytic rate in response to M8 (a RIG-I agonist), and that glycolysis was an essential requirement for DC activation[35]. We suspect that Mito-located MAVS was participated in this process. Nevertheless, further studies are needed.

Studies exploring these questions would help clarify how glucose metabolism and innate immunity interact. Although more studies are needed to circumvent these limitations, our findings reveal regulatory mechanisms between energy metabolism and antiviral responses for creating opportunities in therapeutic applications for virus infection treatment.

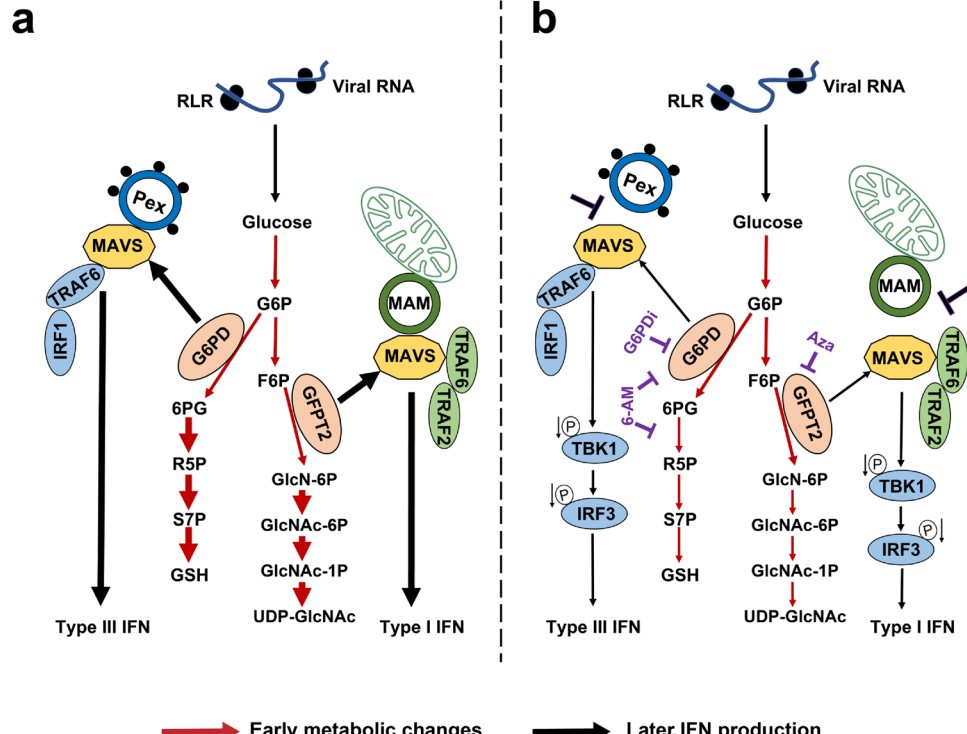

**Fig. 9 | A hypothetical model for differential MAVS placement regulating metabolism and innate immunity.** In RLR signaling, MAVS translocated to peroxisomes and recruits G6PD, leading to the activation of the PPP. Then, TRAF6 and IRF1 interact with MAVS initiating signaling cascades that lead to the production of type III IFN. Conversely, MAVS translocated to MAMs and recruits GFPT2, leading to the activation of the HBP. Then, TRAF6, and TRAF2 interact with MAVS, leading to the production of type I IFN (**a**) (Fig. 9a). When PPP and HBP are inhibited by drugs (**b**) (Fig. 9b), phosphorylation of TBK1 and IRF3 downstream of MAVS is inhibited, resulting in a decrease of the respective IFN responses. Red arrows indicate early metabolic changes. Black arrows indicate later IFN production. The thickness of the arrow represents the enhancement or weakening of the reaction.

## Methods

### Ethics statement
All animal experiments were performed in accordance with the National Institutes of Health Guide for the Care and Use of Laboratory Animals. The protocol was approved by the institutional animal care and use committee of Wuhan University.

### Cells and viruses
All cells were purchased from the American Type Culture Collection (ATCC). Human lung epithelial cells (A549), Human embryonic kidney cells (HEK293T) were cultured in Dulbecco's modified Eagle's medium (DMEM) (Gibco-BRL, Gaithersburg, MD, USA). THP-1 cells were cultured in RPMI 1640 medium (Gibco-BRL, Gaithersburg, MD, USA). BMDMs were cultured in DMEM containing 10% heat-inactivated fetal bovine serum (FBS; Merck Millipore, MA, USA) and M-CSF (10 ng/ml). All cells supplemented with 10% FBS, 100 U/ml penicillin (Gibco, Gaithersburg, MD, USA), and 100 U/ml streptomycin sulfate at 37 °C in 5% $CO_2$. Vesicular stomatitis virus (VSV) and Sendai virus (SeV) were provided by Mingzhou Chen of Wuhan University.

### VSV plaque assays and VSV challenge in vivo
The supernatants were collected and diluted to $10^{-6}, 10^{-5}, 10^{-4}, 10^{-3}$ and $10^{-2}$ with DMEM and used to infect confluent Vero cells (China Center for Type Culture Collection, CCTCC) cultured in 24-well plates. One hour later, the cells were washed with PBS twice and cultured in a mixture of warm 3% low melting point agarose and DMEM containing 10% FBS, 1% methylcellulose and 1% streptomycin and penicillin for 72 h. Cells were stained with 0.2% crystal violet for 2 h, and the overlays were removed. The numbers of plaques were counted, averaged and multiplied by the dilution factor to determine the viral titer (PFU/ml).

Mice between the ages of 8–10 weeks were infected with VSV by a single intraperitoneal injection. Animal survival was monitored for 12 days post VSV injection. Serum and tissues including spleen, liver and lungs were collected at 24 h post injection for immunological and histological analyses. A multiplex ELISA was performed on supernatants using ELISA kits (R&D Systems). Note that IFN-λ was analyzed by an IFN-λ (IL-28B) ELISA (R&D Systems), but we found that this ELISA is largely cross-reactive to IFN-λ (IL-28A) and thus were not able to differentiate between these two mice IFN-λ.

### Mice
The $Mavs^{-/-}$ mice were previously described and kindly provided by Dr. Hong-Bing Shu (Wuhan University). C57BL/6 mice were purchased from the Center for Animal Experiment of Wuhan University. Male and female mice were sex-matched and used at 8-12 weeks of age. All mice were housed in the specific pathogen-free animal facility at Wuhan University.

### Antibodies and reagents
Anti-Phospho-IκBα (Ser32) (2598 s), anti-IκBα (4812 S), anti-GFPT2 (6917) and anti-GAPDH (97166), anti-β-tubulin (86298), anti-ubiquitin (3936), anti-G6PD (12263) were purchased from Cell Signaling Technology (Beverly, MA, USA). Anti-β-actin (ab179467), anti-Myc (ab32) were purchased from Abcam. Anti-HA(H6908) and anti-Flag (M2) were purchased from Sigma (St. Louis, MO, USA). Anti-TRAF6 (sc-7221) and anti-MAVS (sc-365333) were purchased from Santa Cruz Biotechnology (Santa Cruz, CA, USA). HRP-conjugated goat anti-mouse IgG, F(ab')2 fragment specific (115-035-006), HRP-conjugated goat anti-rabbit IgG, F(ab')2 fragment specific (111-035-006) were purchased from Jackson Immuno Research. Neutralizing antibodies against IFNα, IFNβ

and IFNλ were purchased from R&D Systems (Minneapolis, USA). G6PDi-1 (SML2980), 6-aminonicotinamide (6-AN) (A68203), azaserine (104981), OSMI-1 (SML1621) were purchased from Sigma-Aldrich (St. Louis, MO, USA). Unless specified otherwise, all biochemical reagents were purchased from Sigma-Aldrich (St. Louis, MO, USA).

### Quantitative real-time PCR

Total RNA was isolated using RNAiso plus (Takara, Tokyo, Japan) according to the manufacturer's instructions. A sample of 2 µg of total RNA was reverse transcribed using random primers and then quantitative PCR assays were performed using the Bio-Rad CFX manager (Bio-Rad). Primers specific to either human or murine genes are listed in Table S1. The relative expression of each gene was calculated and normalized using the $2^{-\triangle\triangle Ct}$ method. The PCR was terminated at the cycle 30 and the products were visualized by agarose gel electrophoresis.

### Plasmids and shRNA

The coding regions of MVAS, TRAF6, TRAF2, G6PD, GFPT2 and mutants, RIG-I, TAK1, NEMO, IKKα, IKKβ, and TRAF3 were created in our laboratory. The IFN-β, IFN-stimulated response element (ISRE), IRF1 and NF-κB luciferase reporter plasmids were provided by Prof. Hongbing Shu (Wuhan University). All constructs were confirmed by DNA sequencing (Sangon Biotech, Shanghai, China). To verify constructs and the specificity of antibodies, all constructs were transfected into 293 T cells, and expression was analyzed using Western blot. ShRNAs used in this study are listed in Table S2. Note that Sigma-Aldrich Corporation has tested those shRNAs efficiency, and we tested shRNAs efficiency in our laboratory again.

### Immunoblot analysis and coimmunoprecipitation

Cells were lysed in lysis buffer containing 50 mM Tris-HCl (pH 8.0), 150 mM NaCl, 1% Nonidet P-40, 0.1% SDS, 2 mM EDTA and protease inhibitors. The protein concentration was determined by Pierce BCA Protein Assay kit (#PI23225, Fisher Scientific, MA, USA). The samples were subjected to 10% or 12% SDS-PAGE and then transferred to PVDF membrane (Millipore, MA, USA). The membranes were blocked with 5% BSA for 1 h at room temperature before incubation with the specific primary antibodies. After three washes, the membranes were incubated with HRP-conjugated secondary antibodies. Proteins were detected with Clarity Western ECL substrate (Bio-Rad, Hercules, CA, USA) and visualized using an LAS-4000 instrument (Fujifilm, Tokyo, Japan).

Coimmunoprecipitation was performed as previously described (1). Briefly, cells were cultured in 6-cm dishes and lysed in 600 µL lysis buffer containing 20 mM Tris-HCl (pH 7.5), 150 mM NaCl, 1 mM EDTA, and 1% Nonidet P-40, 0.1% (v/v) of a protease inhibitor mixture (Merck, MA, USA), followed by centrifugation at 13,680 × g for 15 min. The supernatants were incubated overnight at 4 °C with 0.5 µg of the indicated antibody cross-linked to 30 µL protein G-agarose. After five washes with lysis buffer, immunocomplexes were resuspended in 20 µL 1x SDS sample buffer for analysis by SDS-PAGE. Due to the close molecular weight of the target proteins or other practical reasons, we have run some of the same samples in different gels/blots and probed them with different antibodies. We have labelled in the source data which blots show the same samples probed with different antibodies.

### Transfection and luciferase reporter gene assays

THP-1 cell were transfected using electroporation with an Amaxa Nucleofector II device according to the manufacturer's protocol. BMDMs were transfected using transfection reagent (#AD600150, Zeta Life, USA). Unless specified otherwise, cells were transfected using Lipofectamine 3000 (Invitrogen, Carlsbad, CA, USA) according to the manufacturer's instructions.

For luciferase reporter gene assays, cells were seeded on 24-well dishes and transfected using methods mentioned above. Twenty-four hours later, cells were serum-starved for an additional 24 before harvest. A Renilla luciferase reporter vector pRL-TK was used as the internal control. Luciferase assays were performed using a dual-specific luciferase assay kit (Promega, Madison, WI, USA). Firefly luciferase activities were normalized on the basis of Renilla luciferase activities.

### Critical commercial assay kits

Mitochondria were isolated by using the Mitochondria Isolation kit (#89874, Thermo, MA, USA) according to the manufacturer's instructions. For hexokinase activity detection, mitochondria were isolated from cells and pellet was lysed and subjected to Hexokinase activity measurement by using Hexokinase Colorimetric assay kit (#K789-100, Biovision, CA, USA). G6PD activity measurement by using G6PD activity assay kit (#MAK015, Sigma-Aldrich, MO, USA). NADPH level performed using NADPH quantification kit (#V9510, Promega, WI, USA). NAD⁺/NADPH level were performed using NADP/NADPH quantification kit (#G9081, Promega, WI, USA). Sccinate levels were measured using a Succinate Assay Kit (#b204718, Abcam) and lactate Colorimetric/Fluorometric Assay kit (#K607-100, Biovision) according to the manufacturer's protocol.

### Subcellular fractionation

MAM, mitochondria, and peroxisome were isolated from cells using Percoll density gradient fractionation as described (2). Equivalent amounts of protein from each fraction were separated by SDS-PAGE and analyzed by immunoblotting.

### Generation of KO cell lines

A549 IFNAR1 KO and IFNLR KO cell line was generated by CRISPR-Cas9 system described before (2). Briefly, a specific oligo targeting the gene was designed using Cas9 target design tools (http://www.genome-engineering.org). The target guide sequence cloning protocol can be found at the Zhang Laboratory GeCKO Web site (http://www.genome-engineering.org/gecko/). HEK-293T cells were co-transfected with the specific lentiCRISPRv2 plasmid, lentivirus packaging plasmid psPAX2, together with envelope plasmid pMD2.G using Lipofectamine 3000. And the lentiviral particles harvested in medium was centrifuged at 15,000 × g for 5 min and then filtered through a 0.22-mm filter (Millipore, MA, USA) to remove cells. When recipient cells were grown to ~70% confluence, they were incubated in fresh culture medium containing 8 mg/ml polybrene. Subsequently, the specific lentiCRISPRv2 lentivirus -containing media was added to the cells. Cells were plated in a 96-wells plate at -1 cell per well to get a single clone. The monoclonal cell colonies were singled out for enlarged culture. KO cell lines were obtained from these enlarged monoclonal cells, and KO was confirmed by Western blotting.

### ¹³Carbon and ¹⁵N-glutamine glucose tracing and steady-state metabolomics

Cells were incubated with $^{13}C_6$-glucose, $^{13}C_{1-2}$-glucose or $^{15}N$-glutamine for 6 h. Mass-spectrometry and metabolite identification were performed on 80% methanol and 20% LC/MS-grade water extracted metabolites. Analyses were performed using a High-Performance Liquid Chromatography and High-Resolution Mass Spectrometry and Tandem Mass Spectrometry (HPLC-MS/MS). The system consisted of a Thermo Q-Exactive in line with an electrospray source and an Ultimate 3000 (Thermo) series HPLC consisting of degasser, a binary pump, and auto-sampler outfitted with an Xbridge Amide column (dimensions of 4.6 mm × 100 mm and a 3.5 µm particle size). The mobile phase A: 20 mM ammonium acetate (pH 9.0), 20 mM ammonium hydroxide, 95% (v/v) water, 5% (v/v) acetonitrile. The mobile phase B: 100% Acetonitrile. The gradient was as following: 15% A (0 min); 30% A (2.5 min); 43% A (7 min); 62% A (16 min); 75% A (15–20 min); 15% A (15–20 min); with a flow rate of 400 µL/min. The capillary of the ESI source was set to 275 °C, with sheath gas at 45 arbitrary units, auxiliary gas at 5 arbitrary units and the spray voltage at 4.0 kV. The top 5 precursor ions were

fragmented using the higher energy collisional dissociation cell set to 30% normalized collision energy in MS2. Data acquisition and analysis were carried out by Tracefinder 2.1 software (Thermo Fisher Scientific) and Xcalibur 4.0 software (Thermo Fisher Scientific).

## Ubiquitination assays

Cells were lysed in buffer containing 30 mM Tris-HCl (pH 7.4), 150 mM NaCl, and 1% NP40 with a cocktail, and the cell lysates were denatured at 95 °C for 5 min in the presence of 1% SDS. A portion of cell lysates were retained for immunoblot analysis to detect the expression of target proteins. The rest of cell lysates were diluted with lysis buffer and immunoprecipitated (Denature-IP) with beads and antibody. The immunoprecipitates were washed three times and subject to immunoblot analysis.

## Immunofluorescence

In brief, Hela cells were plated in 14-mm confocal dishes, transfected with indicated plasmids and treated with virus. After infection, cells were fixed with 4% paraformaldehyde for 15 min and permeabilized with PBS containing 0.2% Triton X-100 for 5 min at room temperature. Samples were blocked with PBS containing 3% BSA for 1 h at room temperature. Then, the cells were immunostained with the indicated primary Abs 1.5 h at room temperature followed by incubation with the relevant dye-conjugated secondary Abs at 37 °C for 1 h. The cells were imaged using a fluorescence microscope (Leica, Germany) with 100× objective lens. The analysis for colocalization was conducted with Image J.

## Seahorse assay

The day before the assay, the Seahorse cartridge was placed in the XF calibrant and incubated overnight. On the day of the assay, cells were seeded into the Seahorse 96-well plate and incubated overnight (Seahorse Bioscience). Then, the media were changed to XF media for 1 h. For the glycolytic capacity, XF Glycolysis Stress Test Kit was used. Glucose, oligomycin and 2-deoxy glucose (2-DG) were diluted into XF media and loaded into the cartridge to achieve final concentrations of 10, 1, and 50 mM, following the standard Seahorse protocol. For cellular mitochondrial function, XF Cell Mito Stress Test Kit was used. Oligomycin, FCCP, antimycin and rotenone were diluted into XF media and loaded into the cartridge to achieve final concentrations of 1, 1, 5, and 1 μM, following the standard Seahorse protocol.

## Metabolomics analysis

The metabolite extraction was performed as previously described (3). Briefly, the media were aspirated, and the cells were washed twice before lysing the cells. The metabolites were extracted using cold 80% methanol/water mixture and resuspended in 50% methanol/water mixture for further analysis using LC-MS/MS. A selected reaction monitoring LC-MS/MS method with positive and negative ion polarity switching on a Xevo TQ-S mass spectrometer was used for analysis. Peak areas integrated using MassLynx 4.1 were normalized to the respective protein concentrations. The data acquisition was carried out using Analyst 1.6 software, and peaks were integrated with MultiQuant (AB SCIEX, Framingham, MA, USA).

## Statistical analysis

Data were obtained from three independent reproducible experiments. Data were expressed as mean ± standard deviations or mean ± the standard error of the mean. Statistical significance was determined using Student's unpaired two-tailed $t$ test, or one-way or two-way ANOVA multiple comparison test as indicated in the legend.

## Reporting summary

Further information on research design is available in the Nature Portfolio Reporting Summary linked to this article.

## Data availability

All data associated with this study are presented within the paper or in the Supplementary materials. Raw data underling the results are provided with this paper. Source data are provided with this paper.

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

## Acknowledgements

This work was supported by the National Key Research and Development Program of China (2021YFC2701800, 2021YFC2701804), the National Natural Science Foundation of China (U22A20335), the Fundamental Research Funds for the Central Universities (2042022dx0003), the Science Fund for Distinguished Young Scholars of Hubei Province (2021CFA054), the Huxiang high-level talents gather engineering innovation and entrepreneurship talents in Hunan province (2021RC5006), the Fundamental Research Funds for the Central Universities (2042021kf023), the National Natural Science Foundation of China (81872262) and Deutsche For schungsgemeinschaft (Transregio TRR60).

## Author contributions

Y.Z. and S.L. conceived and designed the experiment. Q.H., Y.H., L.N., F.D., Z.K. and Q.Z. performed the experiments. L.Z., H.C., Q.W. and F.W. analyzed the data. H.R. and H.Y. processed and typeset the figures. K.X. and Z.L. wrote the manuscript. All authors read and approved the final manuscript.

## Competing interests

The authors declare no competing interests.
