## [Peer Review File · Nature Communications]

MAVS integrates glucose metabolism and RIG-I-like receptor signalingREVIEWER COMMENTS

Reviewer #1 (Remarks to the Author):

This manuscript by He et al reports that RLR activation drives a switch from glycolysis to the PPP and the HBP pathways using the antiviral adaptor protein MAVS. The differential location of MAVS either at the peroxisomes or at MAMs dictates a glucose shift towards the PPP of HBP, respectively. Additionally, through the association with different signaling proteins and transcription factors, MAVS coordinates a specific type I or type III IFN response which overall demonstrates that MAVS integrates glucose metabolism and RIG-I like receptor signaling to regulate IFN responses.

Overall, this study represents an interesting and relevant observation in the field of immunometabolism and displays a set of nicely executed experiments. The total body of experimental data is convincing. The article is correctly written, and statistics appropriately applied throughout the manuscript. Material and methods are also sufficiently described for others to be able to reproduce the main findings of the study. Overall, the manuscript by He et al could be of interest for the broad readership of Nature Communications, however there are some critical points that should be addressed by the authors in a revised version of their manuscript

- The manuscript is rather logical in presentation and correctly written, however it is extremely dense in data presentation and overwhelming at some point. I would highly encourage the authors to submit a stripped-down version of their manuscript by including less data and focusing on what they think are the key findings needed to support their conclusions. Additionally, the panels are extremely small, and it is almost impossible to see/read the content of each panel/figure. This should also be improved. Finally, for transparency, each histogram chart should display each individual data points.
- In general, this article has lots of co-IPs (some have better quality than others) and also present important novel findings on the positioning of MAVS at different locations that dictates its interactions with different binding partners. The important findings should be backed up using confocal microscopy or another flow-based technique.
- Can the authors comment on the choice of using A549 cells to overexpress G6PD? A549 cells are known to display NRF2-addiction which confers them with natural high G6PD and PPP levels. Additionally, NRF2 addiction also alters IFN pathways in general. Probably not the best model to study the relevance of their findings.....

- How do the authors integrate the work recently published by Zevini et al (Front Cell Infect Microbiol) in their concept? The findings of this manuscript should be approached in the discussion of this manuscript.

Reviewer #2 (Remarks to the Author):

In this manuscript the authors show that VSV infection alters cellular metabolism leading to an increase in the first steps of glycolysis but a decrease in later steps of glycolysis. This is explained by an increase in the Pentose Phosphate Pathway (PPP) and the Hexosamine biosynthesis pathway (HBP) which would take away the early glycolytic intermediates from glycolysis. Interestingly, when MAVS is knocked out, they find an increase in all steps of glycolysis but not PPP or HBP indicating MAVS plays a role in the alteration of the utilization of glycolytic intermediates. They go on to show that VSV induced MAVS localizes to the peroxisome and the MAM. Localization to the peroxisome leads to interactions with G6PD of the Pentose phosphate pathway and induction of type III interferons. Localization of MAVS to the MAM leads to interactions with GFPT2 of the HBP and type I interferons. They also use a mouse model of VSV infection to show that inhibition of these metabolic pathways in vivo leads to increased viral replication, presumably due to a decrease in IFN response due to the metabolic block. Overall, the data are largely convincing and is very comprehensive, the in vivo data nicely supports the cell culture data and they draw important conclusions in regard to the interactions of the innate immune system through MAVS activation, metabolic alterations and interferon induction. However, a few clarifications and controls are still needed in some places. The overall conclusions are highly interesting and the model in Figure 9 summarizes the findings. However, they could go further in this model by ordering the reliance of the metabolic changes and interferon induction based on their findings rather than putting them all side by side.

Major points:

1. In figure 1 and S1 they show C-13 glucose tracing. In S1 they show the amount of labeled M+2 pyruvate changes but not M+1. However, from C-13 (1,2) glucose, pyruvate gets a different labeling profile of M+2 if glycolysis is the main pathway and M+1 if the PPP is the main pathway. Because they show a decrease in glycolysis and an increase in the PPP, I would expect increases in M+1 in figure S1. This could be explained by the PPP being used for nucleotides and not shunted back to pyruvate, but this not discussed. This should be more clearly described in the results section.
2. Supplemental figure S3 shows that increase in GPT2 mRNA is time dependent and increase does not occur until late but the metabolic changes ascribed to HBP happen early in figure 1. Do the changes require GPT2 upregulation or other modifications that don't rely on upregulation.
3. For Figures 5A and 5G the mouse experiments are missing a critical control of 6-AN or Aza alone without infection to determine if it is deadly to mice. This might be found in the literature so may not need to be done here but should be described.

4. Figure 6A is confusing, IP with Flag to bring down GFPT2 and blot with HA to see if TAK1, MAVS, TRAF6, NEMO or IKKb come down. All lanes have a band but MAVS and TRAF6 have extra band. What is the changing lower band in all lanes? What are the expected sizes of everything? This panel is not explained clearly enough.

5. In Figure 8F where are G6PD and GFPT2 in the absence of MAVs? They are not in cytosol, mito, MAMs or Peroxisomes. So where would they be?

6. Figure 9 shows the conclusions of the findings. However, Figure S7 shows overexpression of GFPT2 or G6PD leads to alterations of VSV replication. This would indicate that the metabolic changes are upstream of IFN induction. Also, OSMI-1 inhibits MAM localized IFN beta induction further indicating that metabolic changes are between MAVs and IFN induction not side by side as indicated. Taking all the data together it might be more helpful to order the metabolic changes and IFN induction for each rather than putting both side by side.

Minor points:

1. In 2H they measure the levels of NADPH (and in S3) but in the figure itself they have NAPDH. It is correctly spelled in the figure legend but not in multiple places in both figure panels.

2. In Fig S3 J-L what is measured? activation of GPT2? The figure legend does not state.

3. Line 193: Figure S4C: They claim that MAVS-WT is in the cytosol but the westerns show that it is on all the membranes but not seen in the cytosol.

4. Figure S5 legend in two places (line 1069 and 1072) STATE WT AND MAVS -/- BMDMS were transfected but it looks like all of these experiments are done in knock out mouse cells there are no wild type

Reviewer #3 (Remarks to the Author):

In the manuscript “MAVS integrates glucose metabolism and RIG-I-like 2 receptor signaling”, the authors report that the protein MAVS regulator glucose uptake and metabolism during infection with viruses that activate RLRs. The role of MAVS on peroxisomes and mitochondria was most notable, as the authors found that mitochondrial MAVS promoted glucose entry into the glycolysis pathway and peroxisomal MAVS directed glucose into the pentose phosphate pathway. These findings dovetail nicely with prior work on how mitochondrial and peroxisomal MAVS display differences in the genes that they induce during viral infections. Overall, this work is well done, unexpected and interesting. Below I offer comments that are designed to improve what is already in a very interesting study.

1. The authors cite several reviews on innate immunity in the introduction that are over 10 years old. Based on the progress made over the last decade in this field, these reviews are outdated. The authors are encouraged to replace these references with more contemporary review articles. Example on line 88.

2. The authors present genetic evidence that MAVS (localized to different organelles) is required for glucose to enter the glycolysis or pentose phosphate pathways (PPP). They also present genetic evidence that these metabolic pathways are required for MAVS-induced inflammatory gene expression. These findings are confusing, as it seems like each cellular process is controlled by the other pathway. It is possible that the metabolic pathways are not actually required for MAVS-induced interferon and cytokine expression, as suggested by the authors. Rather, the metabolic pathways may be necessary to fuel the continuous expression of genes that is necessary to achieve wild type levels of interferon and cytokine expression. The authors should test this hypothesis by examining early events that occur downstream of MAVS signaling, such as the activation of TBK1 and IRF3. If the activation of these factors is unaffected by metabolic disruptions, then I would suggest that glycolysis and PPP are needed to fuel continuous MAVS-dependent gene expression.

Reviewer #1:

This manuscript by He et al reports that RLR activation drives a switch from glycolysis to the PPP and the HBP pathways using the antiviral adaptor protein MAVS. The differential location of MAVS either at the peroxisomes or at MAMs dictates a glucose shift towards the PPP of HBP, respectively. Additionally, through the association with different signaling proteins and transcription factors, MAVS coordinates a specific type I or type III IFN response which overall demonstrates that MAVS integrates glucose metabolism and RIG-I like receptor signaling to regulate IFN responses.

Overall, this study represents an interesting and relevant observation in the field of immunometabolism and displays a set of nicely executed experiments. The total body of experimental data is convincing. The article is correctly written, and statistics appropriately applied throughout the manuscript. Material and methods are also sufficiently described for others to be able to reproduce the main findings of the study. Overall, the manuscript by He et al could be of interest for the broad readership of Nature Communications, however there are some critical points that should be addressed by the authors in a revised version of their manuscript

Response: Thank you so much for the good words about our work!

(1) The manuscript is rather logical in presentation and correctly written, however it is extremely dense in data presentation and overwhelming at some point. (i) I would highly encourage the authors to submit a stripped-down version of their manuscript by including less data and focusing on what they think are the key findings needed to support their conclusions. (ii) Additionally, the panels are extremely small, and it is almost impossible to see/read the content of each panel/figure. This should also be improved. (iii) Finally, for transparency, each histogram chart should display each individual data points.

Response: Thank you very much for this exciting suggestion. (i) Partial datas were removed to supporting figures. (ii) The panels were enlarged according to the suggestion. If some panels still blurry, please don't hesitate to contact me. (iii) All histogram charts were revised according to the suggestion.

(2) In general, this article has lots of Co-IPs (some have better quality than others) and also present important novel findings on the positioning of MAVS at different locations that dictates its interactions with different binding partners. The important findings should be backed up using confocal microscopy or another flow-based technique.

Response: Additional experiments were performed according to the suggestion. As shown in new Supplementary Figure 14, virus infection forced MAVS and GFPT2 to translocate to MAM. And, VSV infection also forced MAVS and G6PD to translocate to Pex.

(3) Can the authors comment on the choice of using A549 cells to overexpress G6PD? A549 cells are known to display NRF2-addiction which confers them with natural high

G6PD and PPP levels. Additionally, NRF2 addiction also alters IFN pathways in general. Probably not the best model to study the relevance of their findings.....

Response: It is true, high G6PD levels were in A549 cells. To address this issue, additional G6PD knockdown experiments were performed. As shown in new Supplementary Figure 9e,f, knockdown of G6PD or GFPT2 enhanced VSV replication accompanied by lower ISG56 expression in A549 cells. According to those results, we believe that high level of G6PD in A549 cells can not affect our conclusion. In addition, most of the results were obtained from BMDMs in this manuscript.

(4) How do the authors integrate the work recently published by Zevini et al (Front Cell Infect Microbiol) in their concept? The findings of this manuscript should be approached in the discussion of this manuscript.

Response: We added a small statement in this respect to the discussion section according to the suggestion.

Reviewer #2:

In this manuscript the authors show that VSV infection alters cellular metabolism leading to an increase in the first steps of glycolysis but a decrease in later steps of glycolysis. This is explained by an increase in the Pentose Phosphate Pathway (PPP) and the Hexosamine biosynthesis pathway (HBP) which would take away the early glycolytic intermediates from glycolysis. Interestingly, when MAVS is knocked out, they find an increase in all steps of glycolysis but not PPP or HBP indicating MAVS plays a role in the alteration of the utilization of glycolytic intermediates. They go on to show that VSV induced MAVS localizes to the peroxisome and the MAM. Localization to the peroxisome leads to interactions with G6PD of the Pentose phosphate pathway and induction of type III interferons. Localization of MAVS to the MAM leads to interactions with GFPT2 of the HBP and type I interferons. They also use a mouse model of VSV infection to show that inhibition of these metabolic pathways in vivo leads to increased viral replication, presumably due to a decrease in IFN response due to the metabolic block. Overall, the data are largely convincing and is very comprehensive, the in vivo data nicely supports the cell culture data and they draw important conclusions in regard to the interactions of the innate immune system through MAVS activation, metabolic alterations and interferon induction. However, a few clarifications and controls are still needed in some places. The overall conclusions are highly interesting and the model in Figure 9 summarizes the findings. However, they could go further in this model by ordering the reliance of the metabolic changes and interferon induction based on their findings rather than putting them all side by side.

Response: We thank the reviewer for the appreciation of our work. Figure 9 were reorganized according to the suggestion.

Major points:

1. In figure 1 and S1 they show C-13 glucose tracing. In S1 they show the amount of labeled M+2 pyruvate changes but not M+1. However, from C-13 (1,2) glucose, pyruvate gets a different labeling profile of M+2 if glycolysis is the main pathway and

M+1 if the PPP is the main pathway. Because they show a decrease in glycolysis and an increase in the PPP, I would expect increases in M+1 in figure S1. This could be explained by the PPP being used for nucleotides and not shunted back to pyruvate, but this not discussed. This should be more clearly described in the results section.

Response: Actually, the amount of labeled M+1 pyruvate was increased in Supplementary figure 1. It is due to the dense in data presentation, and we apologize for not being more clear. Glucose tracing metabolomics datas were changed to another type of histogram.

2. Supplemental figure S3 shows that increase in GFPT2 mRNA is time dependent and increase does not occur until late but the metabolic changes ascribed to HBP happen early in figure 1. Do the changes require GFPT2 upregulation or other modifications that don't rely on upregulation.

Response: Thank you very much for bringing this issue up. Additional experiments were performed to test whether GFPT activity was regulated. Indeed, GFPT activity was induced as early as 1 hours after VSV infection (new Supplementary Figure 3m). Thus, we suspect that virus regulate GFPT mediated metabolic changes by two ways. At the early stages, virus induce GFPT activity. At the late stages, virus induce the expression of GFPT2 mRNA.

3. For Figures 5A and 5G the mouse experiments are missing a critical control of 6-AN or Aza alone without infection to determine if it is deadly to mice. This might be found in the literature so may not need to be done here but should be described.

Response: Thank you very much. The datas of 6-AN or Aza alone without infection were provided in new Figure 5a,e.

4. Figure 6A is confusing, IP with Flag to bring down GFPT2 and blot with HA to see if TAK1, MAVS, TRAF6, NEMO or IKK β come down. All lanes have a band but MAVS and TRAF6 have extra band. What is the changing lower band in all lanes? What are the expected sizes of everything? This panel is not explained clearly enough.

Response: Thank your for point it out. Lower band in all lanes is IgG heavy chain. We apologize for not being more clear. Figure 6a were reorganized according to the suggestion.

5. In Figure 8F where are G6PD and GFPT2 in the absence of MAVS? They are not in cytosol, mito, MAMs or Peroxisomes. So where would they be?

Response: Thank your for point it out. Figure 8F were reorganized according to the suggestion. G6PD and GFPT2 were located in cytosol in the absence of MAVS (Figure 8F).

6. Figure 9 shows the conclusions of the findings. However, Figure S7 shows overexpression of GFPT2 or G6PD leads to alterations of VSV replication. This would indicate that the metabolic changes are upstream of IFN induction. Also, OSMI-1 inhibits MAM localized IFN beta induction further indicating that metabolic changes

are between MAVs and IFN induction not side by side as indicated. Taking all the data together it might be more helpful to order the metabolic changes and IFN induction for each rather than putting both side by side.

Response: Good suggestion. Figure 9 were reorganized according to the suggestion.

Minor points:

1. In 2H they measure the levels of NADPH (and in S3) but in the figure itself they have NAPDH. It is correctly spelled in the figure legend but not in multiple places in both figure panels.

Response: We apologize for this typo. We changed figures according to the suggestion.

2. In Fig S3 J-L what is measured? activation of GFPT2? The figure legend does not state.

Response: Thank you very much. The figure legend of Fig S3 were reorganized according to the suggestion.

3. Line 193: Figure S4C: They claim that MAVS-WT is in the cytosol but the westerns show that it is on all the membranes but not seen in the cytosol.

Response: Sorry, we are confused. MAVS-WT is in the cytosol in Figure S4C. We apologize for not following your suggestion.

4. Figure S5 legend in two places (line 1069 and 1072) state WT AND MAVS^{-/-} BMDMs were transfected but it looks like all of these experiments are done in knock out mouse cells there are no wild type.

Response: The figure legend of new Figure S6 were reorganized according to the suggestion.

Reviewer #3:

In the manuscript “MAVS integrates glucose metabolism and RIG-I-like receptor signaling”, the authors report that the protein MAVS regulator glucose uptake and metabolism during infection with viruses that activate RLRs. The role of MAVS on peroxisomes and mitochondria was most notable, as the authors found that mitochondrial MAVS promoted glucose entry into the glycolysis pathway and peroxisomal MAVS directed glucose into the pentose phosphate pathway. These findings dovetail nicely with prior work on how mitochondrial and peroxisomal MAVS display differences in the genes that they induce during viral infections. Overall, this work is well done, unexpected and interesting. Below I offer comments that are designed to improve what is already in a very interesting study.

Response: We thank the reviewer for their appreciation of our work.

1. The authors cite several reviews on innate immunity in the introduction that are over 10 years old. Based on the progress made over the last decade in this field, these reviews are outdated. The authors are encouraged to replace these references with more

contemporary review articles. Example on line 88.

Response: Thank your very much. Old references are updated according to the suggestion.

2. The authors present genetic evidence that MAVS (localized to different organelles) is required for glucose to enter the glycolysis or pentose phosphate pathways (PPP). They also present genetic evidence that these metabolic pathways are required for MAVS-induced inflammatory gene expression. These findings are confusing, as it seems like each cellular process is controlled by the other pathway. It is possible that the metabolic pathways are not actually required for MAVS-induced interferon and cytokine expression, as suggested by the authors. Rather, the metabolic pathways may be necessary to fuel the continuous expression of genes that is necessary to achieve wild type levels of interferon and cytokine expression. The authors should test this hypothesis by examining early events that occur downstream of MAVS signaling, such as the activation of TBK1 and IRF3. If the activation of these factors is unaffected by metabolic disruptions, then I would suggest that glycolysis and PPP are needed to fuel continuous MAVS-dependent gene expression.

Response: Sorry, we are confused. To the best of our knowledge, if glycolysis and PPP are needed to fuel continuous MAVS-dependent gene expression, the downstream of MAVS signaling should be affect by metabolic disruptions. Additional experiments were performed to test this hypothesis. As shown in new Supplementary Fig. 7c, metabolic disruptions inhibit the phosphorylation of TBK1 and IRF3.

REVIEWERS' COMMENTS

Reviewer #1 (Remarks to the Author):

The authors have satisfactorily addressed all the comments initially raised.

There are two other minor points that the reviewer would like to raise :

- a careful check should be made on the English of the response to the reviewers' comments. There are a lot of mistakes that should be fixed since this part of the work will also be published online along with the rest of the manuscript

- the authors should provide all the raw data from the immunoblotting analysis in a supplementary figure

Reviewer #2 (Remarks to the Author):

The authors have adequately responded to all this reviewer's comments. However, as raised in the first review, the discussion and figure 9 do not appear to fully address the conundrum of order of activation. It is clear that MAVs localization to the peroxisome leads to activation of type III IFN with activation of the PPP, while MAVs localization to the MAM leads to activation of type I IFN response and of HBD. However, they show that drug inhibition of the PPP or HBD leads to inhibition of the respective IFN responses. The connection not shown in the figure 9 model slide is how inhibition of the PPP or HBD would lead to inhibition of the IFN response. As they show data for this connection, it should be better explained in the discussion.

Reviewer #1 (Remarks to the Author)

The authors have satisfactorily addressed all the comments initially raised.

There are two other minor points that the reviewer would like to raise:

1- a careful check should be made on the English of the response to the reviewers' comments. There are a lot of mistakes that should be fixed since this part of the work will also be published online along with the rest of the manuscript

Response: Thank you for your careful review and constructive suggestions regarding our manuscript. We apologize for the errors in our previous response and have now corrected them.

2- the authors should provide all the raw data from the immunoblotting analysis in a supplementary figure

Response: We have uploaded the raw data of immunoblotting analysis before. It may be that the data cannot be displayed due to a conversion error in the PDF and we have re-uploaded the data.

Reviewer #2 (Remarks to the Author)

The authors have adequately responded to all this reviewer's comments. However, as raised in the first review, the discussion and figure 9 do not appear to fully address the conundrum of order of activation. It is clear that MAVs localization to the peroxisome leads to activation of type III IFN with activation of the PPP, while MAVs localization to the MAM leads to activation of type I IFN response and of HBP. However, they show that drug inhibition of the PPP or HBD leads to inhibition of the respective IFN responses. The connection not shown in the figure 9 model slide is how inhibition of the PPP or HBP would lead to inhibition of the IFN response. As they show data for this connection, it should be better explained in the discussion.

Response: Thank you for your comments and providing good suggestions for our manuscript. We have revised discussion and Figure 9 in manuscript according to your suggestion and new additions are indicated in orange.